# Spatial oxidation of L-plastin downmodulates actin-based functions of tumor cells

Emre Balta [1], Robert Hardt [2], Jie Liang [1], Henning Kirchgessner [1], Christian Orlik [1], Beate Jahraus [1], Stefan Hillmer [3], Stefan Meuer[4], Katrin Hübner [1], Guido H. Wabnitz [1] & Yvonne Samstag [1]

Several antitumor therapies work by increasing reactive oxygen species (ROS) within the tumor micromilieu. Here, we reveal that L-plastin (LPL), an established tumor marker, is reversibly regulated by ROS-induced thiol oxidation on Cys101, which forms a disulfide bridge with Cys42. LPL reduction is mediated by the Thioredoxin1 (TRX1) system, as shown by TRX1 trapping, TRX1 knockdown and blockade of Thioredoxin1 reductase (TRXR1) with auranofin. LPL oxidation diminishes its actin-bundling capacity. Ratiometric imaging using an LPL-roGFP-Orp1 fusion protein and a dimedone-based proximity ligation assay (PLA) reveal that LPL oxidation occurs primarily in actin-based cellular extrusions and strongly inhibits cell spreading and filopodial extension formation in tumor cells. This effect is accompanied by decreased tumor cell migration, invasion and extracellular matrix (ECM) degradation. Since LPL oxidation occurs following treatment of tumors with auranofin or γ-irradiation, it may be a molecular mechanism contributing to the effectiveness of tumor treatment with redox-altering therapies.

[1] Section of Molecular Immunology, Institute of Immunology, Heidelberg University Hospital, 69120 Heidelberg, Germany. [2] Mass Spectrometry Core Facility, Center for Molecular Biology, Heidelberg University, 69120 Heidelberg, Germany. [3] Electron Microscopy Core Facility, Heidelberg University, 69120 Heidelberg, Germany. [4] Institute of Immunology, Heidelberg University Hospital, 69120 Heidelberg, Germany. Correspondence and requests for materials should be addressed to Y.S. (email: yvonne.samstag@urz.uni-heidelberg.de)

ROS are physiologically produced in response to several stimuli, such as cytokines and growth factors, both in tumor cells[1] and in immune cells[2]. Their major cellular sources are NADPH oxidases (NOXes) and the incomplete reduction of oxygen to water in the mitochondrial electron transport chain system. Elevated ROS levels have been observed in several pathophysiological conditions, such as malignant disorders. Tumors have a unique redox homeostasis characterized by a shift towards a pro-oxidative state. To avoid the detrimental effects of ROS, tumor cells evolve to cope with oxidative stress primarily by upregulating antioxidant genes and diminishing ROS generation[3]. In this regard, several studies showed that both the glutathione and TRX1 systems protect tumor cells from deleterious effects of ROS and contribute to tumor initiation and progression[4,5]. Inhibition of TRX1 with PX-12 (1-methylpropyl 2-imidazolyl disulfide) or of TRXR1 with auranofin has been found to effectively diminish tumor growth[6,7]. The effects of such inhibitors are currently being evaluated in various clinical trials. However, the molecular targets of TRX1 and TRXR1 contributing to these effects remain largely elusive.

Post-translational modifications (PTMs) are fundamental for cellular adaptation and rapid cellular responses to changing microenvironments[8]. Among the PTMs, thiol oxidation has long been considered to be an important regulator of cellular functions under physiological and pathophysiological conditions. Thiol oxidation of cysteines in various proteins regulates the subcellular localization, structure, and function of these proteins. In this context, several protein tyrosine phosphatases[9,10], cell cycle regulatory proteins[11,12], growth factors, and actin cytoskeleton-regulating proteins are known to be regulated by thiol switches[13–15]. Increasing evidence suggests that spatiotemporal regulation of ROS in different cellular compartments may play physiological roles through cysteine thiol oxidation[16,17]. Therefore, spatial regulation of thiol switches may also contribute to the polarized and dynamic spatial rearrangement of the actin cytoskeleton.

Tumor progression and metastasis require a highly dynamic and polarized actin cytoskeleton for tumor cell migration and invasion into tissues. The actin cytoskeleton is remodeled by actin-binding proteins, enabling the formation of specialized cellular structures such as filopodia, lamellipodia, stress fibers, and invadopodia[18]. One of these key actin-remodeling proteins is LPL. It is physiologically expressed by hematopoietic cells or ectopically (i.e., endogenously but nonphysiologically) expressed by tumor cells of nonhematopoietic origin when they are malignantly transformed. Thus, LPL serves as an important tumor marker[19–21]. As actin polymerization occurs, LPL intercalates into newly synthesized actin filaments and stabilizes these structures[22,23]. In this sense, LPL is a critical regulator of cell–cell interactions, integrin-based adhesion, chemokine-induced polarization, and leukocyte migration[24–27]. We have shown that the ectopic expression and phosphorylation of LPL in human melanoma cells or prostate cancer cells enhances their migration and invasion in vitro[21] and leads to an enhanced metastatic capacity of these tumor cells in vivo[21,28]. Recently, LPL was shown to enable the plasticity of invadopodial and filopodial structures, which is potentially critical for the rapid remodeling of the cytoskeleton in response to stimuli[29].

Given the preferential shift towards a pro-oxidative state in tumor cells and the relevance of the actin cytoskeleton to tumor cell behavior, we investigated redox modifications of LPL and their consequences for tumor cell functions. Our results demonstrate that spatial oxidation of LPL and reversal of this process by TRX1 critically regulate the actin-based cellular functions of LPL in tumor cells.

## Results

**Differential alkylation suggests oxidation on LPL cysteines.** For identification and biochemical characterization of LPL thiol switches, we performed differential alkylation experiments. Of note, since LPL is physiologically expressed in peripheral blood T-cells (PBTs), which are highly sensitive to oxidative stress, we used PBTs for the initial biochemical assays. Lysates of control or $H_2O_2$-treated cells were differentially alkylated using N-ethyl maleimide (NEM) in the first alkylation step, dithiothreitol (DTT)/Tris(2-carboxyethyl)phosphine (TCEP) for the reduction and methyl-PEG-maleimide (mmPEG$_{24}$) in the second alkylation step (Fig. 1a). Binding of mmPEG$_{24}$ to LPL was analyzed by immunoblotting. A molecular shift of 2.6 kDa (molecular weight (MW) of two mmPEG$_{24}$) was observed when the cells were treated with $H_2O_2$ prior to lysis and alkylation (Fig. 1b, lane 6, 7). This finding strongly suggested existence of redox-sensitive cysteines on LPL.

**Cys42 and Cys101 are redox-sensitive cysteines of LPL.** To determine which cysteines on LPL are oxidized and how great the extent of oxidation in cells is, quantitative mass spectrometry (MS) was used. Briefly, lysates of $H_2O_2$-challenged or untreated PBTs were serially labeled with NEM, reduced with DTT and labeled with heavy isotope-coded NEM (d5-NEM). Thereafter, the samples were run on SDS gels, and Coomassie-stained bands corresponding to LPL were analyzed by MS. d5-NEM-labeled peptides indicated cysteine oxidation. Using this technique, we found that Cys101 and Cys42 were strongly alkylated with d5-NEM and that the signal intensity increased with increasing $H_2O_2$ concentrations (Fig. 1c–e, Supplementary Fig. 1a–b). The trace of d5-NEM on the cysteines is shown in the MS/MS spectra of the Cys101- and Cys42- containing peptides with colored arrows (Fig. 1d, Supplementary Fig. 1c–d). The degree of oxidation reached up to 85% ($84.5 \pm 8,9$) using 1 mM $H_2O_2$ (Fig. 1e). Importantly, pre-reduction of the oxidized cysteines with DTT prior to the first alkylation completely reversed the oxidation on Cys42 and Cys101, suggesting reversible disulfide bridge formation (Fig. 1e). Consistently, only minor sulfonylation on Cys42 and Cys101 was observed (Fig. 1f). Moreover, oxidation of three other cysteines (Cys283, Cys336, and Cys460) was detected at very low levels (Supplementary Table 1). Together with the fact that DTT specifically reduces disulfide bridges, our results indicate that $H_2O_2$ treatment led to the formation of a reversible disulfide bridge between Cys42 and Cys101.

**Cys101 oxidation leads to thiol exchange reactions with TRX1.** Next, to investigate which antioxidant system could regulate LPL thiol switches, we used the TRX1 trapping approach. Moreover, we extended our studies to LPL-expressing tumor cell lines. The TRX1 trapping approach is based on the finding that mutated thiol-dependent oxidoreductases lacking the C-terminal cysteine of the CxxC motif (CxxS mutants) form long-lived mixed disulfide intermediates with target proteins. Thus, the target proteins remain covalently linked to the mutant oxidoreductase, which can be immunoprecipitated and used for further analysis[30]. Using this technique, we captured oxidized LPL in PBTs and in various tumor cell lines, such as the PC3 and HEK293 lines, after treatment with sublethal concentrations of $H_2O_2$ (Fig. 2a, b, Supplementary Fig. 2a). To substantiate the findings of the MS experiments, we mutated the highly and weakly oxidized cysteines of LPL to alanine using site-directed mutagenesis. Wildtype (wt) or Cys-Ala mutants of LPL were then transiently expressed in LPL-negative HEK293 cells. Subsequently, TRX1 trapping was performed. The results clearly showed that mutating Cys101 to Ala completely abolished trapping by TRX1, whereas

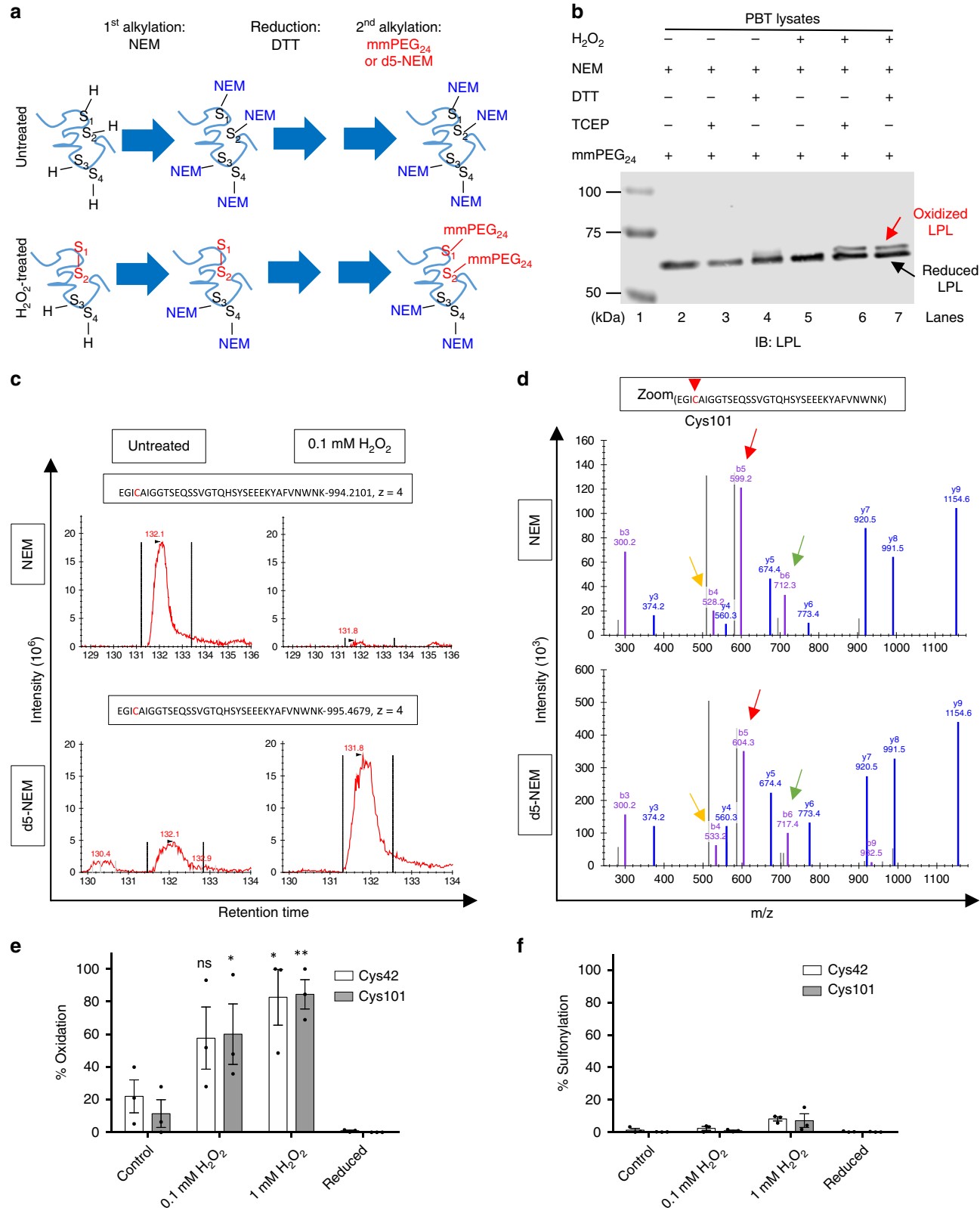

mutating either Cys42 or the other low-oxidized cysteines to Ala did not alter trapping compared to wt LPL (Fig. 2b). This finding strongly suggests that Cys101 is the main redox-sensitive cysteine within LPL and that TRX1 binds to it when it is oxidized.

To substantiate this finding, differential alkylation experiments were performed on wt and C42A- or C101A-recombinant LPL.

LPL was clearly oxidized upon $H_2O_2$ treatment and became accessible to mmPEG$_{24}$ after reduction with DTT (Fig. 2c, lane 7, upper band). Interestingly, without pre-reduction, wt-recombinant LPL existed in a partially oxidized state even in the absence of $H_2O_2$ (Fig. 2c, lane 3, upper band). Consequently, the following experiments were performed after pre-reduction of the

**Fig. 1** Quantitative assessment of LPL oxidation. **a** Workflow of differential alkylation. PBTs were treated with the indicated concentrations of $H_2O_2$ or kept untreated before being subjected to sequential alkylation reactions. For differential alkylation, a first alkylation by NEM, a reduction by DTT/TCEP and a second alkylation by mmPEG$_{24}$ (or d5-NEM) were performed sequentially under denaturing conditions. **b** Differential alkylation of LPL revealed redox regulation on two cysteines upon $H_2O_2$ treatment. Differentially alkylated lysates were run on 6% SDS-polyacrylamide gels and immunoblotted for LPL. Shown is a representative immunoblot (IB) ($n = 3$). **c-f** Quantitative analysis of LPL oxidation by LC/MS-MS. **c** Representative extracted-ion chromatograms (EICs) showing the retention time versus intensity for the LPL peptide, aa 97–131, NEM-labeled (upper graphs) and d5-NEM-labeled (lower graphs) in the absence (left) and presence (right) of 0.1 mM $H_2O_2$. The EICs are based on the summed MS1 intensities of the first three isotopes of the respective peptide ions. **d** Representative MS2 spectrum of the NEM- and d5-NEM-labeled peptide aa 97–131 of $H_2O_2$-treated samples. The colored arrows indicate ions that carry distinct NEM label types (yellow: b4 ion, red: b5 ion, green: b6 ion). The panel depicts magnified spectra (300-1100 m/z range). **e** Quantification of the percent oxidation on Cys42 and Cys101 was performed by dividing the MS1 intensity of the d5-NEM peptide by the intensity of all other forms of the same peptide. The data are presented as the mean ± SEM ($n \geq 3$; *$p < 0.05$, **$p < 0.01$, ns nonsignificant). $P$-values were calculated by two-way ANOVA. **f** The percent sulfonylation on Cys42 and Cys101 was calculated by dividing the MS1 intensity of the sulfonylated cysteine-containing peptide by the total intensity of the same peptide ($n = 3$)

recombinant proteins. Contrarily to wt LPL, C101A, and C42A LPL were not bound by mmPEG$_{24}$ (Fig. 2d). Overall, lack of the MW shift after $H_2O_2$ treatment for C42A LPL and C101A LPL suggests formation of an intramolecular disulfide bridge between these cysteines.

To assess the lowest exogenous $H_2O_2$ levels required to oxidize LPL within cells, PC3 prostate cancer cells were incubated with $H_2O_2$ levels down to 1 μM, and kinetic TRX1 trapping was performed. Twenty micromolar $H_2O_2$ was sufficient to induce substantial oxidation of LPL in PC3 cells (Fig. 2e). Importantly, in untransformed PBTs, which have a lower antioxidant capacity than tumor cells, LPL oxidation was detectable after incubation with as low as 1 μM $H_2O_2$ (Fig. 2f).

**TRX1 knockdown increases LPL oxidation in tumor cells**. To clarify whether the TRX1 system regulates thiol switches of LPL in viable cells, we used PC3 cells. In contrast to T-cells, tumor cells produce considerably higher amounts of TRX1[31,32] that may counteract LPL oxidation. Therefore, upon downregulation of the endogenous TRX1, more LPL should be oxidized and precipitated by the TRX1 trapping mutant (Fig. 3a). To test this hypothesis, TRX1 was knocked down in LPL-expressing PC3 cells using the most effective shRNA, shTRX1_3 (Fig. 3b). Indeed, the amount of trapped LPL in the TRX1 knockdown cells was increased compared to that in the control shRNA-expressing PC3 cells, indicating increased LPL oxidation (Fig. 3c and Supplementary Fig. 2b–c). To independently confirm this finding, wt LPL was expressed in LPL-negative MV3 melanoma cells, and endogenous TRX1 was knocked down with TRX1-specific siRNA (Supplementary Fig. 2d). Proceeding TRX1 trapping in TRX1 knockdown MV3 cells yielded higher amounts of trapped LPL compared to the control siRNA-treated cells (Fig. 3d, Supplementary Fig. 2e–f). These findings imply that the oxidation state of LPL in tumor cells is regulated by TRX1.

**Effects of redox-altering tumor therapies on LPL oxidation**. Upregulation of antioxidant systems, such as the TRX1 system, protects tumor cells from cytotoxic levels of protein oxidation. Given the results presented in Fig. 3b–d, we wondered whether redox-altering tumor therapies lead to LPL oxidation. Thus, we first treated cells with the potent TRXR1 inhibitor auranofin prior to TRX1 trapping. Auranofin treatment of PC3 cells resulted in trapping of LPL even without exogenous $H_2O_2$ (Fig. 3e, Supplementary Fig. 2g–h). These results imply that the TRX1 system is responsible for countering LPL oxidation in tumor cells.

Some conventional anticancer regimens, such as chemotherapy or radiation therapy, act, at least in part, by increasing ROS production in tumor cells[33,34]. Accordingly, we found that γ-irradiation of PC3 cells, ectopically expressing LPL, or of MV3 cells-expressing FLAG-tagged wt LPL indeed led to LPL oxidation (Fig. 3f, g). Altogether, our findings proved that LPL is a highly

redox-sensitive protein that is easily oxidized in response to $H_2O_2$ or current antitumor therapies.

**Actin-bundling capacity of LPL is attenuated by $H_2O_2$**. To delineate whether redox regulation of LPL alters its functions, we next performed an actin cosedimentation assay. Wt-recombinant LPL was mixed with G-actin to allow actin bundle formation, which were then separated from G-actin and actin fibers by low-speed centrifugation[35]. After the optimal LPL/G-actin ratio for this experiment was determined to be 1:1 (Supplementary Fig. 3a–b), the influence of a pro-oxidative micromilieu on the actin-bundling capacity of LPL was tested. For this, wt LPL was preincubated with 50 μM or 200 μM $H_2O_2$. The oxidant was completely removed using catalase (CAT) prior to mixing with actin (Supplementary Fig. 3c–e), since actin would otherwise be oxidized as well. As shown in Fig. 4a, b, oxidation of LPL clearly decreased its actin-bundling capacity in a dose-dependent manner, as indicated by a decrease of actin in the pellets. Furthermore, addition of TRX1 to 50 μM $H_2O_2$-pretreated LPL rescued the function of wt LPL (Fig. 4b). Accordingly, the amount of LPL in the pellet remained unchanged in presence of recombinant TRX1 (Fig. 4c).

**C101A mutation rescues the actin-bundling activity of LPL**. We next compared the actin-bundling capacities of wt LPL, C101A LPL, and C42A LPL under pro-oxidative conditions (Fig. 4d, e). While the percentage of bundled actin decreased dramatically when wt LPL was oxidized by $H_2O_2$, the $H_2O_2$-pretreated C101A LPL mutant showed no significant decrease in its actin-bundling capacity. Similarly, the actin-bundling function of C42A LPL was minimally influenced by $H_2O_2$ treatment (Fig. 4d, e). Altogether, these results indicating the recovery of actin bundling in a pro-oxidative environment either via mutation of LPL to C101A LPL or C42A LPL or via addition of TRX1 provide first biochemical insights into redox-regulated changes in LPL functions.

Note that the actin-bundling function of LPL (Fig. 4d, e), but not the mere binding of LPL to F-actin was impaired under pro-oxidative conditions (Supplementary Fig. 3f–g). To determine the actin-binding capacity of LPL, after coincubation of G-actin and LPL, actin bundles and actin filaments were separated from the G-actin monomers by high-speed centrifugation. $H_2O_2$ treatment did not alter the amount of actin or LPL in the pellet fraction. Thus, these data suggest that oxidized LPL is still able to bind to F-actin.

The effects of LPL oxidation on actin-bundling were further confirmed by transmission electron microscopy (TEM). With this method, actin bundles (Supplementary Fig. 3h, right) could clearly be distinguished from actin filaments (Supplementary Fig. 3h, left). While filamentous actin was observed as thin, wavy structures (diameters 7–15 nm), actin bundles appeared as rod-like structures (thicknesses 23–50 nm). $H_2O_2$ pretreatment of wt

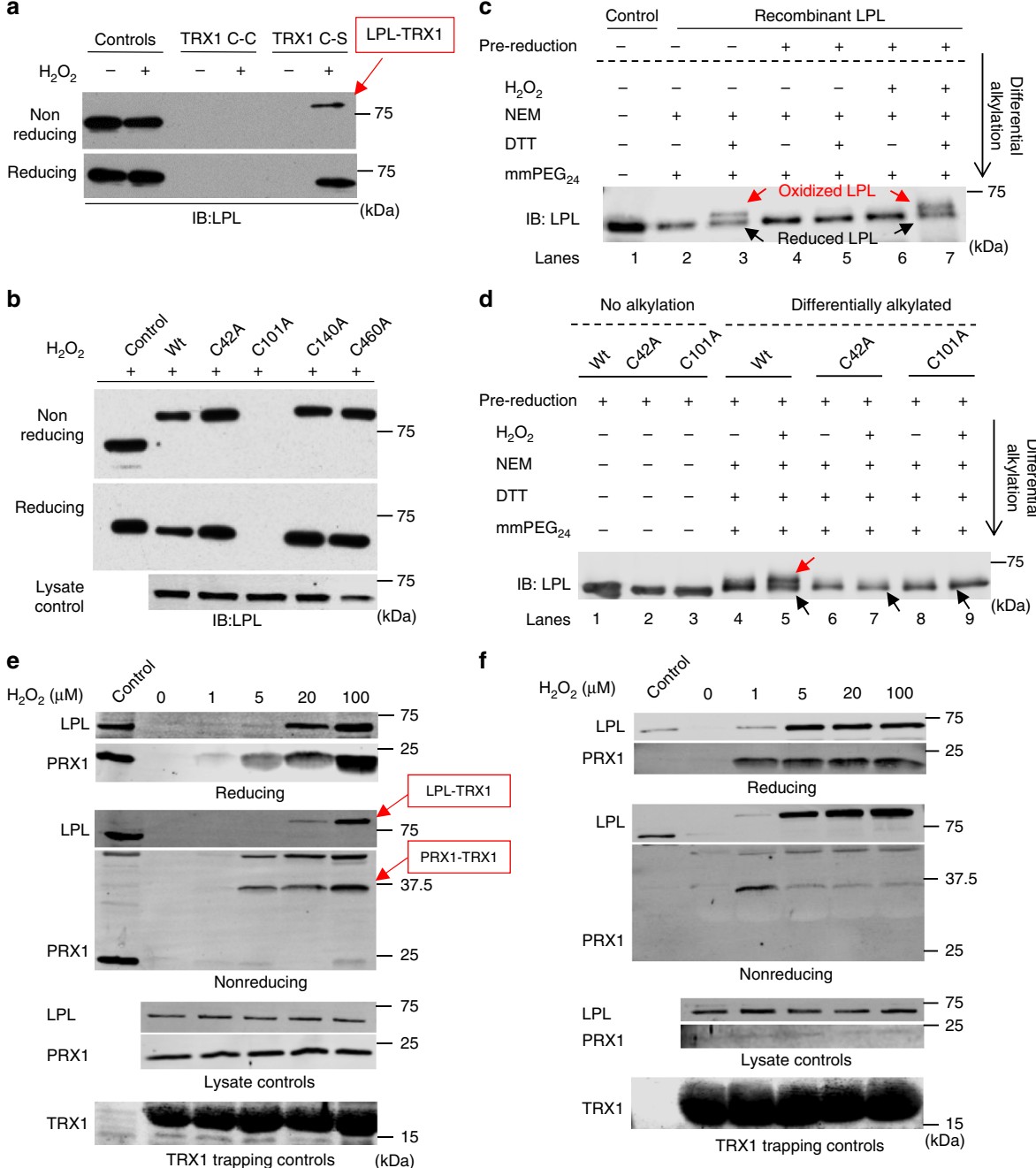

**Fig. 2** TRX1 trapping and differential alkylation of LPL Cys-Ala mutants. **a** Representative immunoblots (IB) showing trapping of LPL by TRX1 C35S upon $H_2O_2$ treatment. PC3 cells were kept untreated or were treated with 0.1 mM $H_2O_2$. Thereafter, the cells were lyzed, and the trapping reaction was performed using TRX1 C35S (the trapping mutant) or TRX1 C35C (wt). TRX1-bound complexes were purified by streptavidin affinity purification and analyzed by western blotting under reducing and nonreducing conditions. Lysates were used as controls, and the other lanes show the trapping by wt TRX1 or C35S TRX1 ($n = 3$). **b** Representative IB of the trapping of different LPL Cys-Ala mutants ($n = 3$). HEK293 cells were transiently transfected with Cys-Ala LPL mutants, and trapping was performed as described. Lysates of LPL-expressing cells were used as controls. **c** Representative IB of differential alkylation of recombinant wt LPL ($n = 3$). Recombinant LPL was directly used or was reduced using DTT prior to differential alkylation. Thereafter, the LPL was kept untreated or was treated with 0.1 mM $H_2O_2$ for 15 min. Following this step, the samples were subjected to differential alkylation. Then, the samples were run on 6% nonreducing or reducing SDS-polyacrylamide gels and immunoblotted for LPL ($n = 4$). **d** Representative IB of differential alkylation of recombinant wt, C42A and C101A LPL after pre-reduction with DTT ($n = 3$). The samples were prepared and immunoblotted for LPL. The red arrow indicates the upper band for the oxidized form of LPL, while the black arrows indicate the lower band for the reduced form of LPL. **e** TRX1 trapping in PC3 cells and **f** PBTs. The cells were treated with the indicated concentrations of $H_2O_2$ and subjected to TRX1 kinetic trapping. Representative IBs stained for LPL and PRX1 are shown ($n = 3$)

LPL led to a dramatic loss of actin bundles (Fig. 4f, left). In contrast, actin bundles were still dominant if C101A LPL mutant was pretreated with $H_2O_2$ (Fig. 4f, right), confirming that LPL oxidation prevents actin-bundling.

**Cellular extensions are sites with high oxidation levels**. ROS, at physiological levels, act spatiotemporally[17]. Therefore, even overall low but locally high concentrations of ROS may induce spatial protein oxidation. Physiologically, locally high

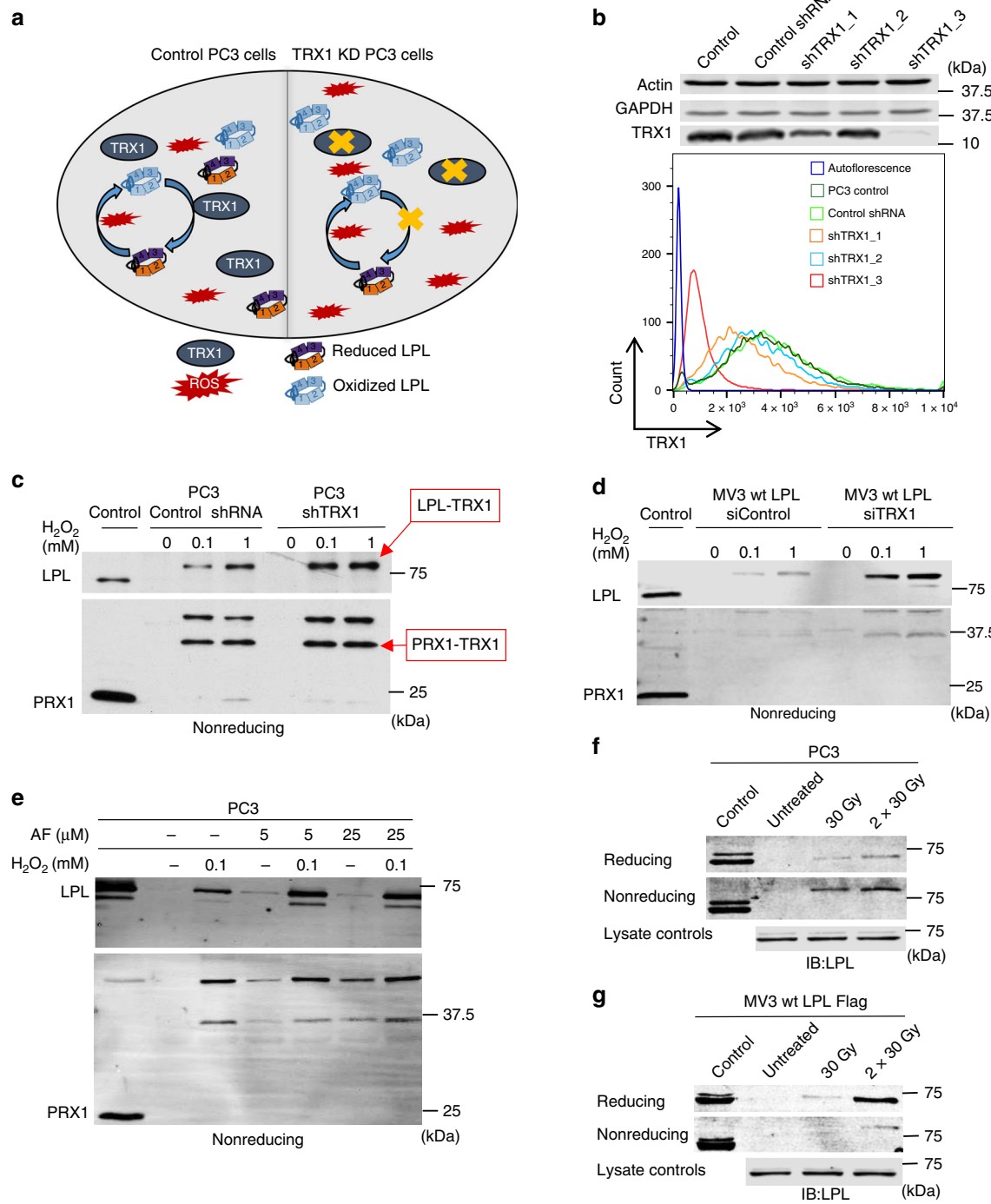

**Fig. 3** TRX1 regulates LPL oxidation in prostate cancer and melanoma cells. **a** Schematic representation of the reduction of oxidized LPL in control (wt) PC3 cells (left) and in TRX1 knockdown PC3 cells (right). **b** Immunoblots (upper panel) and flow cytometry histograms (lower panel) showing stable knockdown of TRX1 in PC3 cells. PC3 cells were transduced with TRX1-targeting shRNAs or control shRNAs ($n = 5$). **c** Immunoblot showing the TRX1 trapping of LPL and PRX1 in control and TRX1 knockdown PC3 cells. The trapping was performed as described. Shown are nonreducing gels ($n = 4$). **d** Immunoblots of TRX1 trapping of LPL and PRX1 in MV3 cells. MV3 cells stably expressing LPL-FLAG constructs were transfected with siTRX1 or control siRNA. Thereafter, trapping was performed as described ($n = 3$). **e** Immunoblot of TRX1 trapping of LPL and PRX1 upon inhibition of TRXR1 in PC3 cells with auranofin ($n = 3$). **f** PC3 cells and **g** wt LPL-FLAG-expressing MV3 cells were either kept untreated or were irradiated with 30 Gy or 60 Gy for 10 min. The cells were then subjected to TRX1 kinetic trapping ($n = 3$)

concentrations may occur through spatially confined ROS production or at locations with low antioxidative protection. Given the regulatory effect of TRX1 on the oxidation state of LPL, we next determined the subcellular localization of LPL, TRX1, and F-actin in MV3 cells expressing eGFP-fused LPL in the presence or absence of $H_2O_2$ (Fig. 5a, b). LPL and F-actin were found in the nucleus, in the cytoplasm and in filopodia of MV3 cells. In contrast, TRX1 was present in the nucleus and cytoplasm, but was absent from filopodia under both control and $H_2O_2$-treated conditions. These data suggest the presence of a pro-oxidative

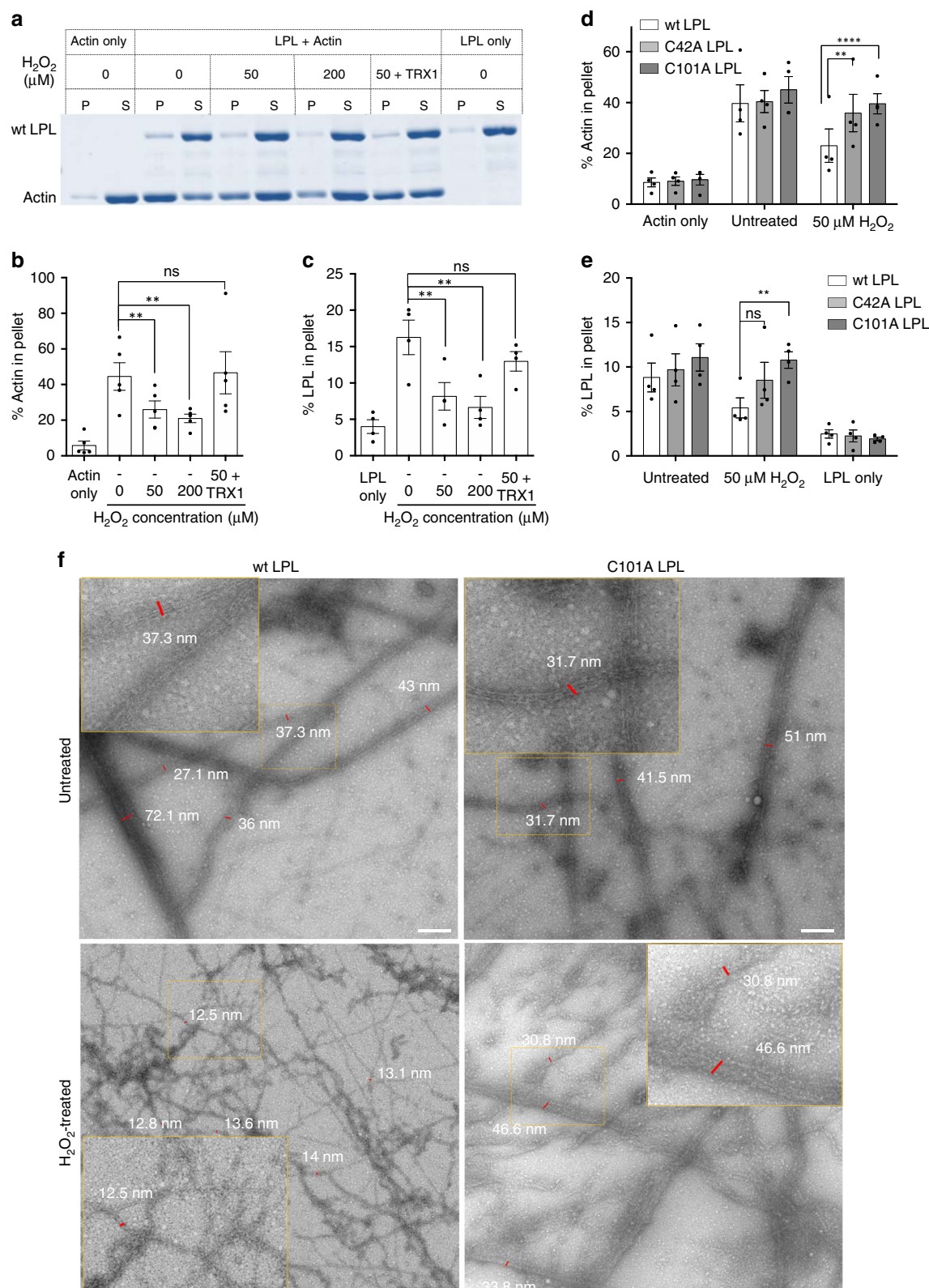

milieu in filopodia that favors the spatial oxidation of TRX1 substrates, e.g., LPL, within these structures. To substantiate this hypothesis, we used the $H_2O_2$ sensor roGFP-Orp1, which allows real-time quantification of intracellular $H_2O_2$[36]. To measure intracellular $H_2O_2$ at sites where LPL is located, we generated an LPL-roGFP-Orp1 fusion cDNA construct and stably expressed it in MV3 cells (Fig. 5c). The fusion of LPL did not

influence the redox potential of roGFP-Orp1 (Supplementary Fig. 4a). Hypothetically, roGFP-Orp1 should show oxidation in cells with high sensitivity, and LPL could provide spatial control because it should mostly be associated with F-actin. Intriguingly, imaging by ratiometric confocal microscopy for up to 1 h revealed higher basal levels of oxidized LPL-roGFP-Orp1 at distal parts of the cells, primarily within filopodial extensions (Fig. 5d, e,

**Fig. 4** C101A LPL and C42A LPL retain their function under pro-oxidative conditions. Actin-bundling capacity of recombinant wt, C101A and C42A LPL under pro-oxidative conditions. Recombinant LPL pretreated with the indicated concentrations of $H_2O_2$ was coincubated with G-actin for 4 h. Actin bundles were separated from F-actin and G-actin by low-speed centrifugation. The pellet and supernatant fractions were loaded onto SDS-polyacrylamide gels and stained with Coomassie Brilliant Blue. **a** Representative Coomassie-stained gel for recombinant wt LPL. P, pellet; S, supernatant ($n = 5$). **b**, **c** Percent quantification of **b** actin and **c** LPL in the pellet fraction. The percentage of protein in the pellet was calculated by taking the ratio of the corresponding signal in the pellet fraction to the total intensity (of supernatant and pellet) ($n = 4$). **d**, **e** Comparison of the actin-bundling capacities of recombinant wt LPL, C101A LPL, and C42A LPL. Quantification of the percentages of **d** actin and **e** LPL in the pellet fraction. The calculations were performed as described above. The data are presented as the mean ± SEM ($n \geq 3$; *$p < 0.05$, **$p < 0.01$, ***$p < 0.001$, ****$p < 0.0001$, ns nonsignificant). P-values were calculated by t-test (**b**, **c**) and by two-way ANOVA (**d**, **e**). **f** Electron micrographs of actin bundles (>20 nm) and actin filaments (<15 nm). Recombinant wt LPL (left) or C101A LPL (right) was kept untreated (upper panels) or pretreated with 50 μM $H_2O_2$ (lower panels) and coincubated with G-actin for 4 h. Thereafter, the samples were prepared and imaged by TEM. Scale bar = 200 nm ($n = 2$)

Supplementary Fig. 4b). Consistently, enhancement of LPL-roGFP-Orp1 oxidation following $H_2O_2$ treatment was more prominent at distal sites (Fig. 5d). This finding suggested that oxidation of LPL was potentially higher within these structures.

**PLA reveals spatial oxidation of LPL**. To determine the spatial oxidation of LPL more directly, we used a dimedone-based PLA that was recently established[17]. Dimedone specifically reacts with cysteine sulfenic acids[37]. To examine spatial LPL sulfenylation, MV3 cells were treated with 100 μM $H_2O_2$, and fixed in the presence of 5 mM dimedone. Consequently, they were immunostained with anti-LPL and anti-dimedone-Cys antibodies and then subjected to PLA. PLA allows the detection of two antibodies that are ~40 nm apart from each other, causing fluorescent dots (puncta) to appear in cells. Indeed, dimedone-LPL puncta were observed predominantly at distal sites of the cells, particularly in cell protrusions (Fig. 6a, Supplementary Fig. 4c). We observed 23 (23.1 ± 1.22) puncta/cell in the absence of $H_2O_2$. The number increased by 44% (to 33.13 ± 1.95 puncta/cell) upon 5 min of $H_2O_2$ treatment (Fig. 6b, c). Consistent with our previous findings, the PLA puncta/cell were significantly lower in C101A LPL-expressing cells than in wt LPL-expressing cells. Moreover, $H_2O_2$ treatment did not further increase the number of PLA puncta in C101A LPL-expressing cells (Fig. 6b, c).

**Involvement of NOXes in spatial LPL oxidation**. In addition to the lack of TRX1 in filopodia, the higher amounts of ROS in the filopodia could also result from local ROS production. In this regard, local activity of NOXes has been proposed to be critical for cell migration and other functions[38,39]. However, a functional NOX inhibitor diphenyleneiodonium (DPI) (Supplementary Fig. 4d) did not significantly influence the basal levels of oxidized LPL-roGFP-Orp1, either in the cytoplasm (Supplementary Fig. 4e) or within filopodia (Supplementary Fig. 4f). Therefore, at least under our experimental conditions, the higher oxidation of the LPL-roGFP-Orp1 sensor in filopodia than in the cell body seemed to occur mostly due to the lack of antioxidant systems.

To more directly address whether NOX activity plays a role in the oxidation of LPL, PLA was performed with LPL-expressing MV3 cells upon DPI treatment. PLA again revealed a significantly higher number of PLA puncta/cell in wt LPL-expressing cells than in C101A LPL-expressing cells in response to $H_2O_2$ treatment (Fig. 6d). Interestingly, DPI-treated wt LPL-expressing cells had fewer PLA puncta per cell than untreated wt LPL-expressing cells, indicating lower spatial oxidation of LPL (Fig. 6d). These differences were not seen in C101A LPL-expressing MV3 cells. Thus, although DPI did not significantly change the oxidation of the sensor in filopodia, spatial oxidation of LPL in wt LPL-expressing MV3 cells, as measured by PLA, could still be related to NOX activity.

Collectively, these results show for the first time, that LPL is oxidized as a result of increased ROS levels due to the activation

of cellular NOXes, the presence of exogenous ROS in the cellular microenvironment, or diminished levels of the TRX1 system. Importantly, this redox regulation of LPL occurs in a spatial manner with a focus at the cell periphery.

**Oxidation of LPL at Cys101 inhibits MV3 cell migration**. In tumor cells, actin-based cell protrusions play a major role in directed cell migration contributing to tumor cell metastasis. Given the high antioxidative capacity of tumor cells and the clinical efforts to treat tumors by antagonizing antioxidants and increasing ROS levels, we next sought to determine whether LPL oxidation plays a role in these processes. To examine this possibility, MV3 cells were chosen since they do not express LPL and overexpression of LPL increased the metastatic capacity of these cells[28]. MV3 cells with similar LPL expression levels were enriched by FACS (Fig. 7a). Consistent with previous findings[28], in the absence of $H_2O_2$, the number of transmigrated cells was significantly higher in LPL eGFP-expressing cells compared to cells expressing eGFP alone. Additionally, mutating the phosphorylation site, Ser5 on LPL to Ala significantly diminished the number of migrated cells, while migration was similar between wt LPL- and C101A LPL-expressing cells (Supplementary Fig. 5a). Thus, we next tested the migration of wt LPL C42A LPL or C101A LPL-expressing cells in a pro-oxidative milieu. $H_2O_2$ treatment interfered with the migration of all MV3 cells (Fig. 7b). Since $H_2O_2$ oxidizes a plethora of proteins, this finding was not unexpected. Nevertheless, despite this relatively strong $H_2O_2$ effect on MV3 cells in general, C101A LPL expression rescued MV3 migration to a low but significant extent.

**LPL oxidation diminishes ECM degradation by tumor cells**. Matrix invasion and degradation via invadopodia formation and chemokine gradient sensing by filopodial structures are strongly dependent on firm and elastic actin bundle formation[40,41]. Therefore, we next investigated cell invasion into the ECM using a Matrigel invasion assay[28]. $H_2O_2$ treatment significantly diminished cell invasion into the Matrigel, which was partially restored by C101A LPL expression (Fig. 7c, Supplementary Fig. 5b). To determine the relevance of LPL thiol switches to matrix degradation, we seeded MV3 cells on fluorescently labeled gelatin in the presence or absence of $H_2O_2$ (Fig. 7d, e). The area of matrix degradation was analyzed by confocal microscopy. Consistently, the matrix degradation capacity of LPL eGFP-expressing cells was stronger than that of MV3 cells expressing eGFP alone. Importantly, under pro-oxidative conditions, C101A LPL-expressing cells showed a stronger matrix degradation capacity than wt LPL- or control eGFP-expressing cells (Fig. 7d, e). Altogether, the results indicate that LPL oxidation on Cys101 downregulates actin-dependent cellular processes such as tumor cell migration, invasion, and matrix degradation.

Degradation of the ECM is achieved via the release and activity of matrix metalloproteinases (MMPs). Thus, the activity of

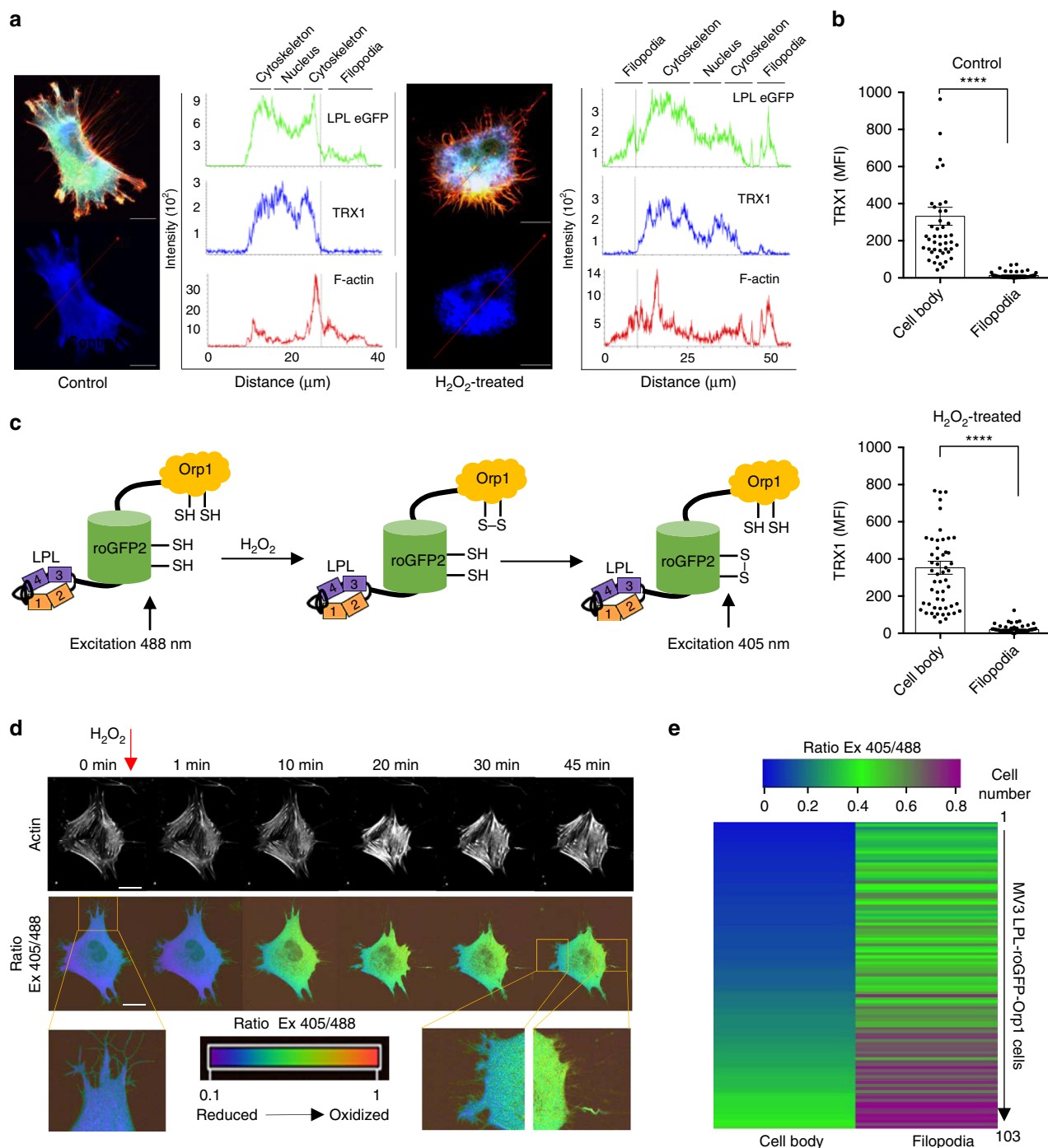

**Fig. 5** LPL-roGFP-Orp1 is spatially oxidized at protrusions of cells. **a** Intensity profiles of TRX1, LPL, and F-actin in untreated (left) and 50 μM $H_2O_2$-treated (right) MV3 wt LPL eGFP cells. MV3 LPL eGFP cells (green) that firmly adhered to poly-D-lysine-coated coverslips overnight were kept untreated or were treated with 50 μM $H_2O_2$ for 30 min. The samples were then fixed and stained for TRX1 (blue) and F-actin (red). The samples were imaged using a confocal microscope ($n = 5$). Scale bar = 10 μm; ×100 magnification. **b** Quantification of TRX1 in filopodia and cell bodies. At least 50 cells were analyzed from three independent experiments. The data are presented as the mean ± SEM ($n \geq 3$; ****$p < 0.0001$, ns nonsignificant). $P$-values were calculated by $t$-test. **c** Schematic diagram of the LPL-roGFP-Orp1 sensor. **d** Representative time-lapse images of MV3 LPL-roGFP-Orp1 cells. The upper panel shows F-actin (white), and the lower panel shows the oxidation state of the roGFP-Orp1 probe. The magnified area shows filopodial extensions. The color scale from dark blue to green and red indicates the oxidation state of the sensor. Ratio imaging was performed with a confocal microscope equipped with dual excitation features ($n \geq 3$). Scale bar = 10 μm; ×100 magnification. **e** Heat map showing the ratio of oxidized (excitation (Ex) 405 nm) to reduced (Ex 488 nm) roGFP-Orp1 in LPL-roGFP-Orp1-expressing cells in the absence of $H_2O_2$ (time-lapse imaging, t = 0 h, three independent experiments, $n = 103$ cells). Each lane represents a single cell. The color scale from blue to green and magenta indicates the oxidation state of the sensor in different cells

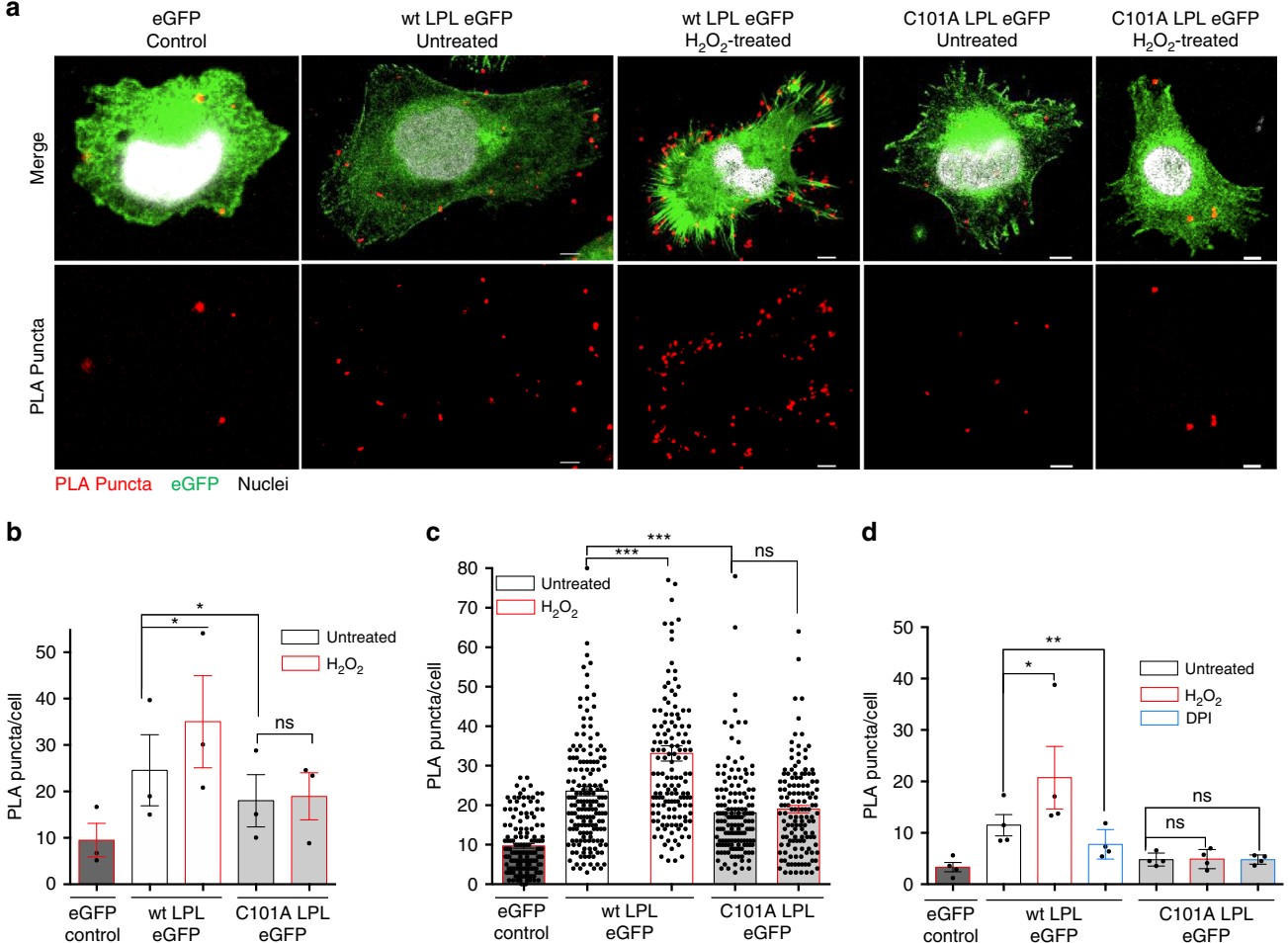

**Fig. 6** Spatial oxidation of LPL at the cell periphery. **a** Confocal microscopy images of MV3 LPL eGFP cells upon PLA staining. Cells were kept untreated or were treated with 100 μM $H_2O_2$. Then, the cells were fixed with PFA in the presence of 5 mM dimedone and stained for LPL and dimedone with specific antibodies and subjected to PLA. Ten xyz optical sections/sample were imaged in each experiment using confocal microscopy. Representative images of eGFP-, wt LPL eGFP-, and C101A LPL-expressing MV3 cells in the absence and presence of $H_2O_2$ are shown. Shown are nuclei (white), LPL eGFP (green) and PLA puncta (red). Each red punctum represents sulfenylation on LPL. Scale bar = 5 μm; ×100 magnification. **b, c** Quantification of the PLA puncta/cell in the presence or absence of $H_2O_2$. **b** Fifty cells were imaged in each of three independent experiments. Each dot in the histograms represents the average PLA puncta/cell for a different experiment. **c** Cumulative comparison of the PLA puncta/cell values in eGFP-, wt LPL eGFP-, and C101A LPL eGFP-expressing MV3 cells in the absence or presence of $H_2O_2$. A total of 150 cells/sample from three independent experiments were imaged. Each dot represents a cell from one of three experiments. **d** Quantification of the PLA puncta/cell in untreated versus DPI- or $H_2O_2$-treated MV3 cells expressing eGFP, wt LPL eGFP, or C101A LPL eGFP. *P*-values were calculated by one-way ANOVA. The data are presented as the mean ± SEM ($n \geq 3$; *$p < 0.05$, **$p < 0.01$, ***$p < 0.001$, ns nonsignificant)

MMPs in the supernatants and the release of specific MMPs into the supernatants of LPL-expressing or control cells were investigated. We could indeed objectify a link between LPL expression and total MMP activity and MMP2 release in MV3 cells (Supplementary Fig. 6a–b). Interestingly, LPL and MMP2 were closely associated with F-actin stress fibers and invadopodial extensions (Supplementary Fig. 6c–d). Moreover, the proteins coimmunoprecipitated, suggesting a close association between MMP2 and LPL (Supplementary Fig. 6e). However, the rescue effects of expression of redox-resistant LPL on these processes were marginal (Supplementary Fig. 6f and Supplementary Note 1).

**LPL oxidation attenuates cellular spreading.** Since LPL is critically involved in the formation of cellular protrusions such as filopodia and invadopodia and is prominently oxidized at the periphery of cells, we next focused on cell spreading and filopodia formation. For this purpose, we used PC3 cells that ectopically express LPL and are adapted to LPL. Endogenous LPL was knocked down in PC3 cells with siRNAs targeting the 3'-UTR of LPL (Fig. 8a). 24 h later, eGFP or LPL eGFP was overexpressed using lentiviral transduction. The knockdown of endogenous LPL and the re-expression of eGFP-tagged LPL constructs were verified by western blotting (Fig. 8b). These PC3 cells were shortly allowed to adhere to coverslips to enable analysis of actin-based processes during initial cell spreading[42]. After that, the cells were kept untreated or were treated with $H_2O_2$ (Fig. 8c, d). Then, spreading of the cells was analyzed (Supplementary Fig. 7a). LPL knockdown cells showed strongly reduced spreading compared to control siRNA-transfected cells (Fig. 8c, e). Wt LPL eGFP and C101A LPL eGFP overexpression reversed this phenotype completely. More importantly, $H_2O_2$ treatment significantly diminished the spreading in wt LPL-expressing cells, while C101A LPL-expressing cells were still able to spread (Fig. 8c, e).

To further elaborate this finding, filopodia were counted (Fig. 8d, Supplementary Fig. 7a–b). MYO10 was used for

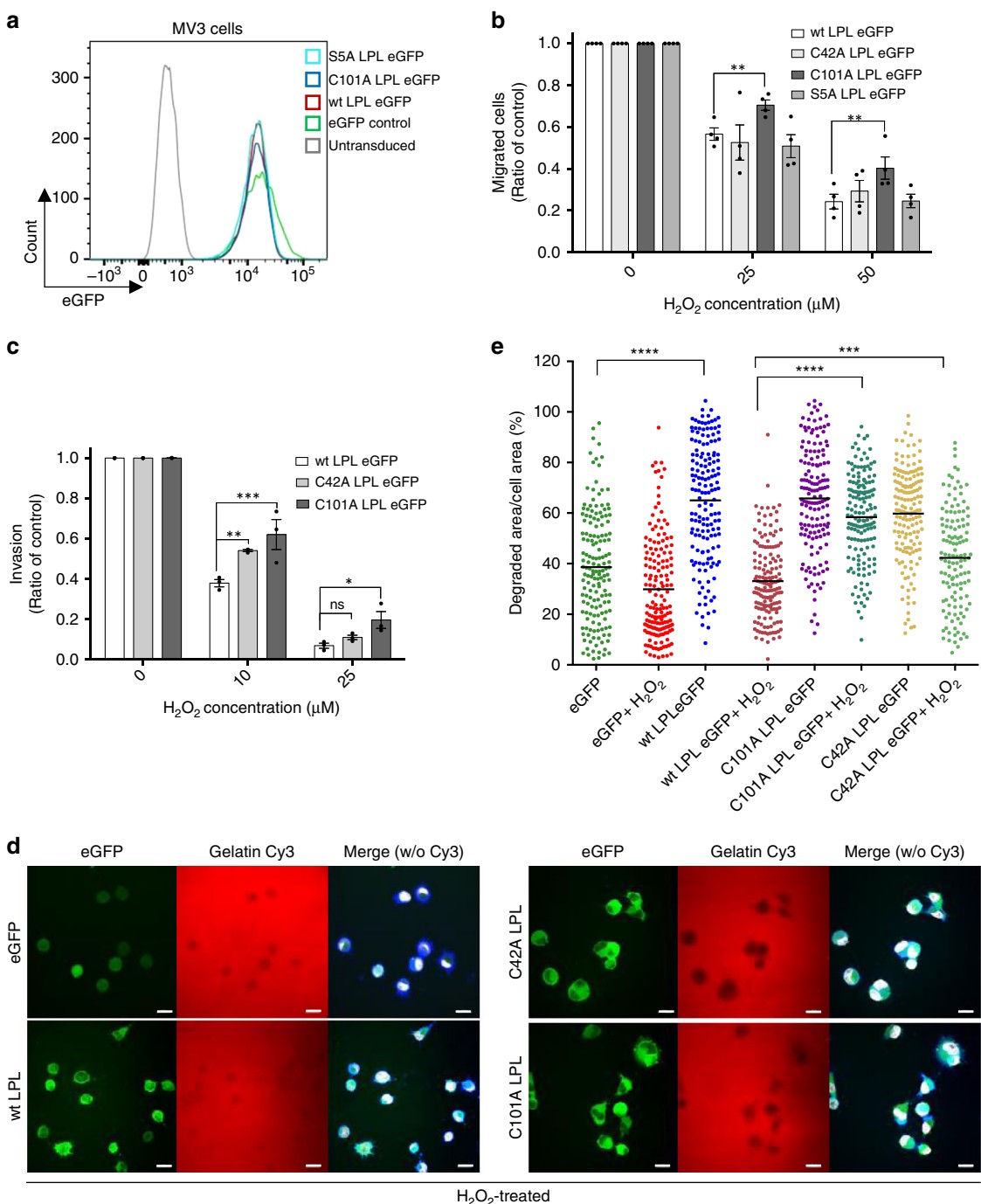

**Fig. 7** The migration and ECM degradation capacities of MV3 cells. **a** Histograms showing stable expression of LPL eGFP in MV3 cells. MV3 cells were transduced with lentiviral vectors expressing the indicated LPL constructs. Cells with intermediate LPL eGFP expression were sorted using FACS. **b** Quantification of the migration of MV3 cells expressing the indicated LPL constructs in 2D Transwell chambers and **c** in 3D invasion chambers. A total of $3 \times 10^4$ cells were added to the upper chamber. Thirty minutes later, the cells were treated with the indicated concentrations of $H_2O_2$. The samples were fixed and mounted on microscopy slides. More than five xy focal planes per sample were imaged using LSM (×20 objective). **d** Representative confocal microscopy images showing the matrix degradation capacity of MV3 cells expressing eGFP alone or eGFP-tagged wt LPL, C101A LPL, or C42A LPL constructs. The cells were allowed to degrade Cy3-gelatin for 3 h in the presence of 25 μM $H_2O_2$. Shown are LPL eGFP (green), gelatin Cy3 (red) and merged images without (w/o) Cy3 (eGFP, F-actin (blue) and nuclei (white)). Scale bar = 20 μm; ×60 magnification. **e** Matrix degradation was quantified as the loss of fluorescence underneath the cells versus the total area of the cells. At least 150 cells were imaged per sample from three independent experiments. P-values were calculated by two-way ANOVA. The data are presented as mean ± SEM ($n \geq 3$; *$p < 0.05$, **$p < 0.01$, ***$p < 0.001$, ****$p < 0.0001$, ns nonsignificant)

quantitative assessment of conventional filopodia because it localizes at the tips of mature filopodia and is critical for cellular events such as migration[43,44]. We found that the number of filopodia was diminished in LPL knockdown cells, which could be rescued by expression of wt or C101A LPL eGFP (Fig. 8f). Consistently, $H_2O_2$ markedly diminished the number of filopodial extensions in the control and wt LPL-expressing cells. In contrast, C101A LPL expression reversed this phenotype.

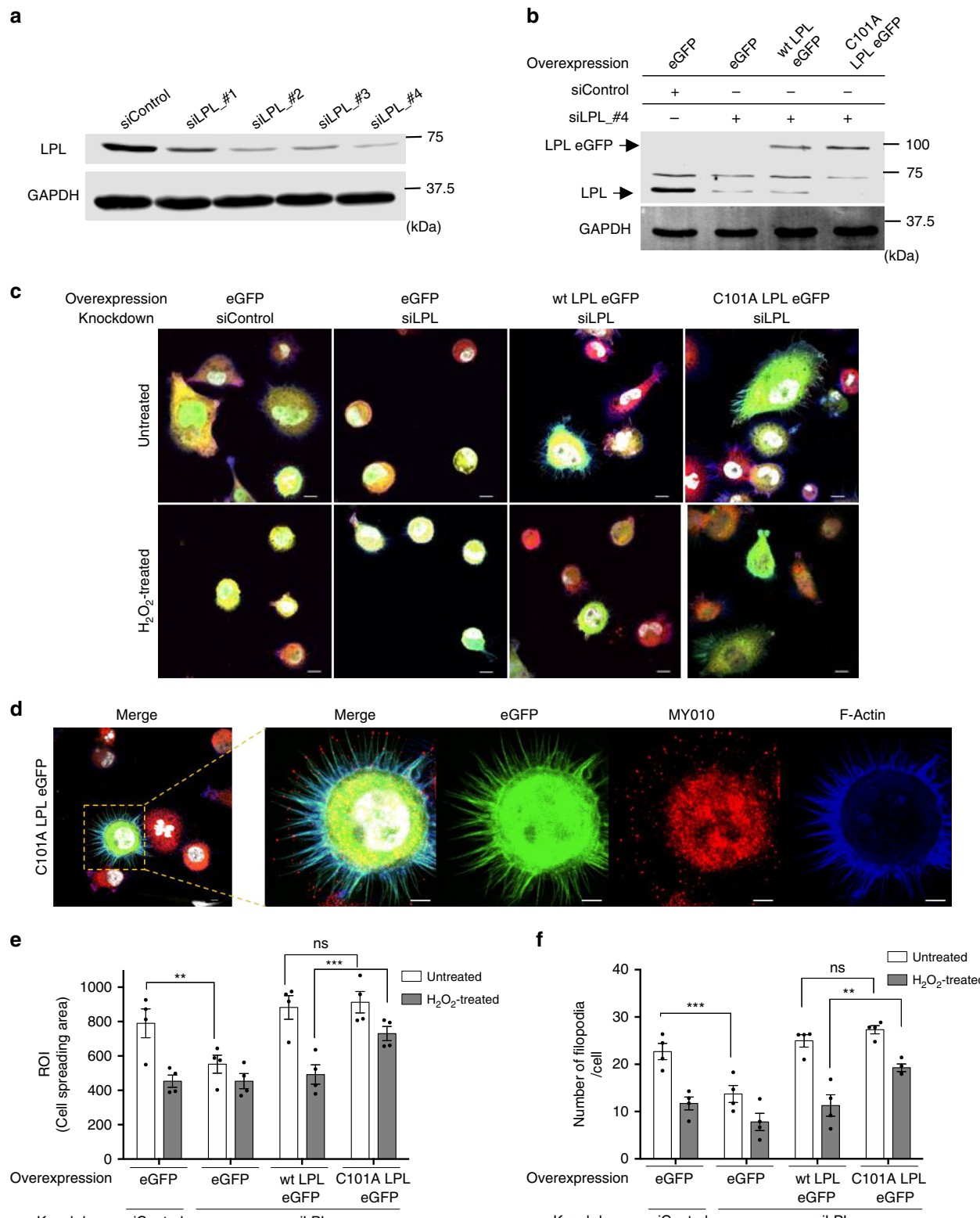

Collectively, these findings revealed a critical involvement of LPL in actin dynamics at the cell periphery that seemed to be attenuated by its oxidation at Cys101.

## Discussion
In this study, we found that reversible oxidation of LPL on Cys101 diminishes its actin-bundling capacity and actin-based cellular functions. While LPL reduction is mediated by the TRX1 system, LPL oxidation is provoked by locally high intracellular ROS levels and preferentially takes place at spatially confined cellular extrusions, such as filopodia and invadopodia. Thus, LPL oxidation likely serves as a molecular switch translating the redox environment into altered functions of tumor cells.

**Fig. 8** LPL oxidation limits spreading and filopodia formation. **a** Representative immunoblot showing siRNA-mediated knockdown of LPL in PC3 cells. Knockdown was assessed three days after siRNA transfection ($n = 3$). **b** Simultaneous knockdown and overexpression of LPL in PC3 cells. The knockdown of endogenous LPL and the overexpression of eGFP-fused LPL were analyzed by western blotting. GAPDH was used as a loading control ($n \geq 3$). **c** Representative confocal images of control (eGFP), wt LPL eGFP-expressing and C101A LPL eGFP-expressing PC3 cells. Cells with simultaneous LPL knockdown and overexpression were seeded on poly-D-lysine-coated coverslips for 30 min to allow weak adherence. Then, the cells were treated with 25 μM $H_2O_2$ (lower panel) or were kept untreated (upper panel). Thereafter, the cells were stained for nuclei (DAPI, white), MYO10 (red), and F-actin (Sir-actin, blue), and confocal images were acquired ($n \geq 4$). Scale bar = 10 μm; ×100 magnification. **d** Magnified image gallery showing LPL expression (eGFP) and MYO10 signals at the tips of filopodia ($n \geq 4$). Scale bar = 5 μm; ×100 magnification. **e** The ROIs of single cells were automatically selected based on eGFP and SiR-actin signals. The calculated areas (ROIs)/cell were used as measures of cell spreading. At least 30 cells were imaged per sample in each of three independent experiments. **f** Quantification of the number of filopodia in untreated or $H_2O_2$-treated cells. To count filopodia, intensity profiles were drawn on the cell extensions. Filopodia were defined by an intensity profile with a two-fold greater mean intensity of MYO10 and SiR-actin than background. At least 25 cells/sample were imaged in four independent experiments. *P*-values were calculated by two-way ANOVA. The data are presented as the mean ± SEM ($n \geq 3$; *$p < 0.05$, **$p < 0.01$, ***$p < 0.001$, ns nonsignificant)

Among the biochemical approaches used to identify redox-sensitive cysteines, quantitative MS analysis provided a strong level of confidence. This degree of certainty was necessary since various pitfalls in redox chemistry led to misidentification of redox-sensitive cysteines on proteins. Such pitfalls include air oxidation or lysis of cellular compartments such as mitochondria[45]. Consistently, although we have identified three other cysteines differentially labeled with d5-NEM, quantification by MS only showed strong and significant oxidation of Cys101 and Cys42. Cys101 oxidation was further confirmed by TRX1 kinetic trapping and differential alkylation experiments with wt LPL versus LPL Cys-Ala mutants. Additionally, the observed molecular mass shift of wt LPL by two mmPEG$_{24}$ molecules was indicative of a disulfide bridge between Cys42 and Cys101. However, the C42A LPL mutant was still trapped by the TRX1 trapping mutant. One plausible explanation for this could be the binding of Cys101 to another cysteine upon oxidation when Cys42 is mutated to Ala. Another explanation could be cyclic sulfenamide formation between sulfenylated Cys101 (C-SOH) and an amide group of a nearby amino acid[10,32].

At the functional level, actin-bundling, but not actin-binding of LPL was attenuated by LPL oxidation. This finding is consistent with proposed models for the actin-binding and actin-bundling function of LPL[46,47]. In their study, Galkin et al. revealed that actin-bundling by the LPL analog fimbrin occurs by sequential binding of the two actin-binding domains (ABDs). The binding of F-actin to ABD2 is required for ABD1 activation and subsequent binding to adjacent F-actin. Interestingly, a recent NMR study further revealed that the binding of ABD1 to adjacent F-actin is regulated by folding of the RH domain[46]. According to that study, folding of the RH domain of LPL onto its EF-hand modules by $Ca^{2+}$ binding could negatively regulate the actin-bundling capacity. Since Cys101 is located in the RH domain and Cys42 is located in the EF-hand module, the disulfide bridge formation described here most likely induced steric inhibition of F-actin binding to ABD1 by stabilizing an RH and EF-hand domain interaction (Fig. 9a). This could explain the loss of actin-bundling without influencing the actin-binding capacity of oxidized LPL.

Tumor cells contact their environment via certain specialized structures. Thus, actin-based cellular extrusions, such as invadopodia and filopodia are pre-requisites for sensing the environment and responding to it. Interestingly, we have shown, for the first time, that such rapidly formed cellular extrusions lack the antioxidant molecule, TRX1. The use of our newly generated ROS sensor LPL-roGFP-Orp1 and a dimedone-based PLA together with 3D confocal imaging revealed that cellular extrusions are spatially restricted sites where high levels of LPL oxidation occurs. Interestingly, global inhibition of NOX activity using DPI

diminished basal levels of LPL oxidation, indicating that NOX activity is involved in LPL oxidation.

Considering the necessity of LPL in cell migration due to its translocation and actin-bundling function at invadopodia, filopodia, and other cellular extrusions[48,49], spatial oxidation of LPL may be a molecular mechanism for the inhibition of tumor cell migration in a pro-oxidative microenvironment. Consistent with this assumption, cell spreading and filopodia formation, in which LPL is crucially involved[29,50], were clearly attenuated in a pro-oxidative environment. Importantly, C101A LPL prevented this effect. Thus, we propose a model in which weak cell adherence is induced after initial contact with the ECM (Fig. 9b, left). Cell spreading is then mediated by actin polymerization at the cell periphery through formation of cellular extrusions (Fig. 9b, middle and right). LPL provides structural elasticity and stability during the formation of such extrusions through its actin-bundling function. In a pro-oxidative milieu, LPL oxidation is more prominent at the cell periphery than in the cell body due to a lack of antioxidants, thereby preventing actin bundling. Therefore, the formation of stable cellular extrusions is impaired (Fig. 9c, d). Consequently, tumor cell migration and invasion can be attenuated at least in part by peripheral oxidation of LPL and prevention of peripheral actin elasticity. Additionally, we have several lines of evidence suggesting a functional link between MMP2 release and LPL. The enhanced MMP2 release by LPL expression, and close association of MMP2 and LPL on F-actin indicate that LPL may help MMP2 translocate to invadopodial structures. However, the influence of LPL oxidation on MMP2 release was weak, which might be attributable to oxidation of other proteins upon exogenous ROS administration and to complex redox regulation of MMPs[51,52]. The interplay of MMPs with actin-bundling proteins need to be further investigated. In conclusion, the enhanced metastasis and invasiveness of LPL-expressing tumor cells may be synergistically mediated by regulation of peripheral actin elasticity and MMP2 transport to invasive sites.

Studying the spatial oxidation of proteins under physiological and pathophysiological conditions, as done here for LPL, provides novel insights into the redox regulation of cellular functions. In this regard, oxidation of actin at methionine residues via MICAL1 has been described[53] and linked to cellular functions[15]. Similarly, compartmentalized ROS production in mitochondria and peroxisomes or in the vicinity of NOXes has been described[38,39]. It was proposed that enrichment of NOX2 at the leading edges of migrating cells is responsible for localized ROS production, which is required for the directional migration of the cells[38,39]. Independently, a recent study revealed spatial oxidation of specific redox-sensitive proteins in the vicinity of NOXes using a dimedone-based PLA approach[17]. Despite these important

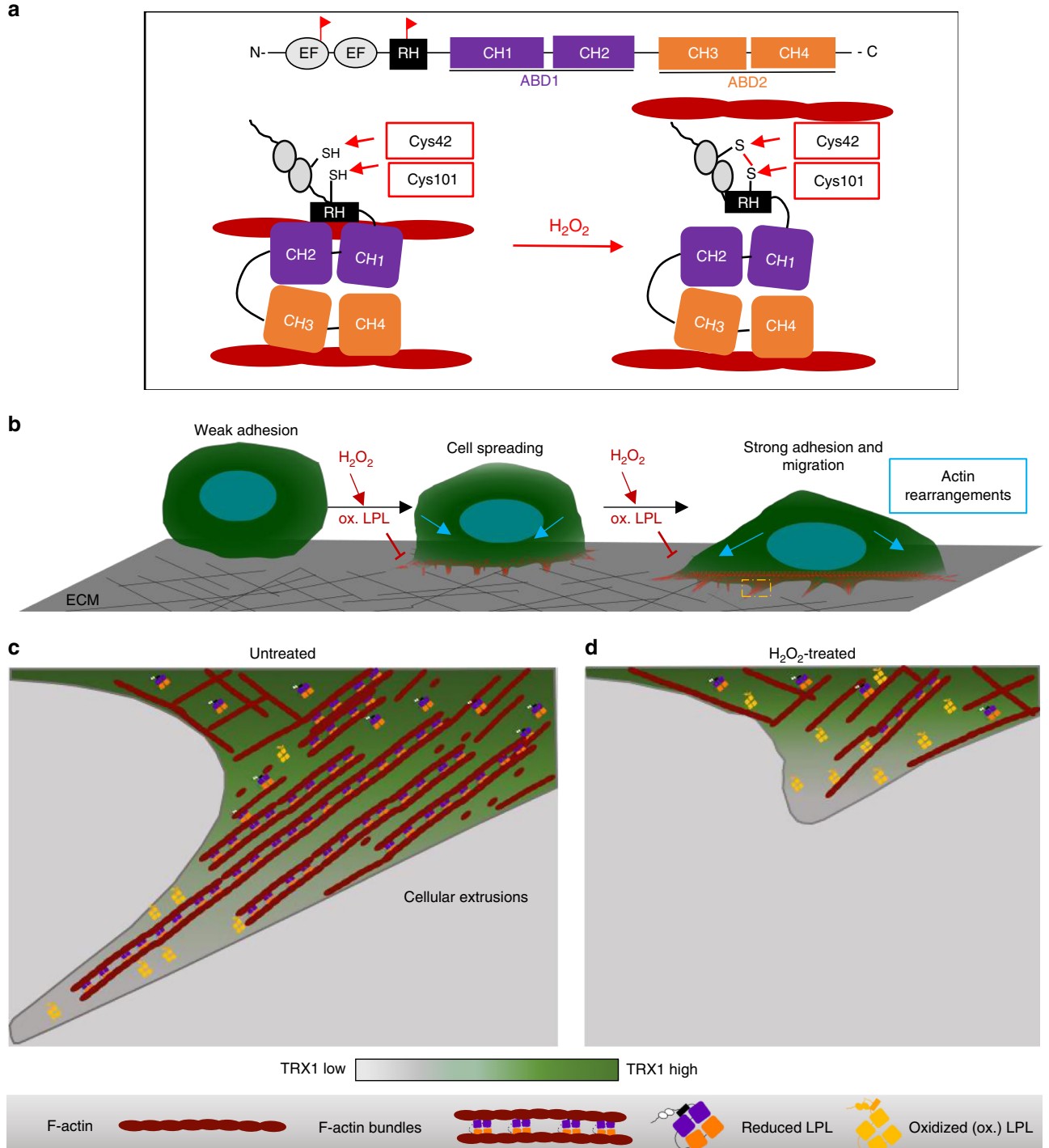

**Fig. 9** Proposed models: **a** the redox regulation of the actin-bundling function of LPL and **b**–**d** the regulation of peripheral actin dynamics by spatial oxidation of LPL. **a** LPL bundles F-actin by sequential binding of ABD1 and ABD2 to adjacent actin filaments. The regulatory helix (RH) domain folds onto ABD2. $H_2O_2$ treatment leads to disulfide bridge formation between Cys101 (located at the RH) and Cys42 (located at the EF-hand module), thereby leading to dissociation of ABD2 or less efficient binding of ABD2 to F-actin. Red colored flags show the location of Cys42 and Cys101 on LPL. **b** Cell spreading through peripheral actin dynamics is halted by $H_2O_2$ treatment. After weak adherence, cells spread by forming actin-based cellular extrusions. Administration of exogenous ROS blocks cellular extrusion formation and cell spreading. **c, d** Enlarged cartoon of the boxed region in **b**. **c** Under control conditions, actin bundling by reduced LPL is involved in the formation of cellular extensions. **d** Under pro-oxidative conditions, the formation of actin bundles by LPL is prevented, impeding the formation of cellular extrusions. The color scale from light to dark green indicates the concentration gradient of TRX1. In the absence of antioxidant systems at such rapidly forming structures, the likelihood of protein oxidation is elevated

findings, there is limited evidence showing spatial oxidation of individual proteins and the relevance of such oxidation to cellular behavior. Our data demonstrate that spatial LPL oxidation within cellular extrusions correlates with low levels of antioxidants and with accumulation of ROS at these specialized structures. The enhanced sensitivity of the cellular extensions to ROS implies that distinct levels of spatial oxidation may play regulatory roles not only in the context of tumors, but also in physiological processes such as dendritic cell migration (by filopodia/dendrites) and T-cell migration. Thus, focusing on spatial oxidation of individual proteins, instead of global oxidation, may provide novel molecular insights into diverse cellular functions.

In various cancer types, high levels of antioxidants have been linked to tumorigenesis[54,55] and chemotherapy resistance[56,57]. Moreover, several chemotherapeutic agents and radiation therapy target tumors by redox modulation[34,57,58]. Interestingly, treatment of tumors with auranofin, an inhibitor of TRXR1 that is currently being tested in clinical trials, resulted in oxidation of LPL, even in the absence of exogenous $H_2O_2$. We further showed that γ-irradiation also led to LPL oxidation. Therefore, LPL oxidation resulting in inhibition of tumor cell migration and invasion may contribute to the inhibitory effects of these therapies on tumor cells.

Importantly, although clinically used TRX1 inhibitors have so far been considered to mainly target tumor cells, there is a risk that such inhibitors also block TRX1 in immune cells, potentially leading to impaired immune responses against tumors. A deeper understanding regarding the redox regulation of proteins in immune cells versus tumor cells needs to be obtained for successful therapies. Oxidation of specific proteins, like LPL, could potentially be used as biomarker both in experimental studies and in clinical trials in which the TRX1 system is targeted by various inhibitors. Finally, to overall address these questions, redox-sensitive in vivo murine tumor models containing a functional immune system need to be established.

## Methods

**Materials**. The following cell culture reagents were purchased: RPMI 1640 (11875093, Thermo Fischer Scientific), DMEM (M-L2624-I, Cell Concepts), fetal calf serum (FCS, Pan Biotech), puromycin (540411, Calbiochem), and L-glutamine (25030081, Life Technologies).

The following materials used in this study were also purchased: All restriction enzymes were purchased from NEB. The plasmids psPAX.2, PMD.2, pLJM1-eGFP, pLKO, and pEGFPN1 were purchased from Addgene according to MTA regulations. A site-directed mutagenesis kit (Invitrogen, A13282) and LPL-specific AcceII siRNAs (A-011716-13 and custom-designed control siRNA and siTRX1, Dharmacon), Vivaspin 6 (VS0601, Sartorius); ZebaSpin desalting columns (89893), BCA protein assay kit (23225), protein concentrators (88513), Pierce centrifuge columns (89868) and Ni-NTA beads (88221), Thermo Fischer Scientific; CAT (C9322), NEM (E3876), mmPEG24 (22713), poly-D-lysine (P6407), Streptavidin sepharose beads (71-5004-40AE), gelatin (G1393), Imidazole (I0250), Biotin (B4501), anti-FLAG M2 affinity gel (A2220), and Duolink In Situ Orange Starter Kit Mouse/Rabbit (PLA, DUO92102) were purchased from Sigma–Aldrich. Additionally, DTT (6908.1, Carl Roth), d5-NEM (D-6141, EQ Laboratories), Polypropylene columns (35964, Qiagen), recombinant actin (AKL99, Tebu-Bio), an actin-bundling assay kit (BK001, Tebu-Bio), a $H_2O_2$ detection kit (ICT-9132, Biomol), 35 mm μ-Dishes (81156, Ibidi), 8-μm pore-sized Transwell inserts (3422, Corning), Matrigel invasion chambers (354480, Corning), a gelatin matrix degradation assay kit (ECM670, Merck), recombinant human TRX1 (ab51064, Abcam), an MMP activity assay kit (ab112146, Abcam), and an MMP antibody array (ab134004, Abcam) were purchased.

Antibodies and fluorescent dyes were purchased following sources and used at the indicated dilutions:: LPL4A.1 mAb (MA5-11921, 1:500), anti-PRDX1 (LF-PA0095, 1:1,000), anti-rabbit AF405 (A-31556, 1:200), and anti-mouse AF405 (A-31553, 1:200) antibodies, Thermo Fischer Scientific; mouse anti-TRX1 (559969, 1:500), anti-CD28 (555725, 5 μg ml$^{-1}$), Annexin-FITC (556547) and 7-AAD (559925), BD Pharmingen; anti-CD3 (OKT3, in-house, 20 ng ml$^{-1}$); anti-MYO10 (22430002, 1:250) and anti-MMP2 (NB200-193, 1:250) antibodies, NovusBio; anti-mouse Cy3 (115-165-146, 1:800), anti-rabbit Cy3 (711-165-152, 1:800), anti-mouse HRP (1:5,000, 111-035-045) and anti-rabbit HRP (1:5,000, 111-035-045) antibodies, Dianova; anti-NOX4 (14347-1-AP, 1:2000) was purchased from Proteintech; SiR-actin (SC001, 200 nM), Tebu-Bio, DAPI (D9542, 1:10,000), and

anti-actin (A5316, 1:2500), Sigma; anti-GAPDH (4300, 0.1 μg ml$^{-1}$), Ambion; anti-GSR (1:2,000, ab16801) was purchased from Abcam; anti-cysteine sulfenic acid (07-2139, 1:600) antibody, Merck–Millipore and anti-mouse IRDye-680RD (926-68072,1:10,000), anti-mouse IRDye-800CW (926-32212, 1:10000), anti-rabbit IRDye-680RD (926-68073, 1:10,000), anti-rabbit-IRDye-800CW (926-32213, 1:10,000), LI-COR Biosciences.

**Cell culture**. Human peripheral blood mononuclear cells (PBMCs) were purified through Ficoll-Hypaque-based density gradient centrifugation of heparinized blood from healthy volunteers upon approval by local authorities. Resting human PBTs were purified via negative magnetic bead selection with a Pan T-Cell Isolation Kit according to the manufacturer's instructions. This study was approved by the Ethics Committee of Heidelberg University (S-089/2015).

The melanoma cell line (MV3) (provided by Dr. van Muijen, University Hospital Nijmegen, The Netherlands) was cultured in RPMI 1640 medium supplemented with 10% FCS. The prostate cancer cell line (PC3) (provided by Dr. M. Cecchini, University of Bern, Switzerland), human embryonic kidney (HEK293) cells and HEK293T (provided by Dr. Steeve Boulant, Heidelberg University) cells were cultured in DMEM (+10% FCS, 4 mM L-glutamine). MV3 and PC3 cell lines were tested by genome sequencing (DSMZ, Braunschweig, Germany). All cells were kept at 37 °C, 5% CO$_2$. For selection of cells stably expressing lentiviral constructs, both PC3 and MV3 cells were cultured in the presence of 1 μg ml$^{-1}$ puromycin.

**Site-directed mutagenesis and cloning**. pEGFPN1-wt LPL vectors was used as the backbone[28]. Site-directed mutagenesis of LPL cysteines to alanine was performed using the site-directed mutagenesis kit. Cloning of FLAG-tagged LPL, eGFP-tagged LPL and GST-tagged LPL were performed using standard restriction-ligation based cloning (Supplementary Table 2).

LPL-roGFP-Orp1 was generated by overlap extension PCR followed by restriction-ligation based cloning. For this, pLCPX roGFP-Orp1, and pEGFPN1-LPL were used as backbones. roGFP-Orp1 and LPL cDNAs were amplified separately followed by overlap extension PCR. Finally, the LPL-roGFP-Orp1 cDNA was cloned into pLJM1 vector.

For TRX1 knockdown experiments, three TRX1-targeting shRNAs and one nontargeting shRNA (control) were designed and cloned into a pLKO.puro vector. The following TRX1-targeting sequences were used for shRNA construct design: shTRX1_1: 5'-GCATGCCAACATTCCAGTTTT, shTRX1_2: 5'-GCAGGTGAT AAACTTGTAG, shTRX1_3: 5'-GCTTCAGAGTGTGAAGTCAAA, and control shRNA: 5'-GGCATTCCAGAGGATGGTAAT.

**Lentivirus production and transduction**. Lentivirus production, titration, and target cell transduction were performed according to standard protocols. For transduction, 1 × 10$^5$ target cells were seeded into 6-well plates. The next day, 100 μl of the virus stock and 8 μg ml$^{-1}$ polybrene were added to the cells. The medium was replaced after 16 h, and cells were selected 48 h after the medium change. Selection of cells with comparable expression was achieved using fluorescence-activated cell sorting (FACS) and 1 μg ml$^{-1}$ puromycin selection.

**Differential alkylation and quantitative mass spectrometry**. PBTs were kept untreated or were treated with 0.1 mM or 1 mM H$_2$O$_2$ for 30 min in RPMI medium supplemented with 0.2% FCS. The cells were then lyzed (1% Triton X-100, 1:100 protease inhibitor cocktail, 0.5 M EDTA and 10 μM NEM in 1x TBS) for 20 min on ice. Cytoplasmic fraction was collected by centrifugation at 10,000 × g for 10 min at 4 °C[28]. As a control for the sensitivity of the method, a completely reduced sample was artificially produced. For this, lysates of 0.1 mM H$_2$O$_2$-treated cells were treated with 20 mM DTT prior to first alkylation. The protein concentration was determined using a BCA kit according to the manufacturer's protocol. Then, 100 μg of each lysate was treated with 100 mM NEM under denaturing conditions (6 M urea, 0.1 M Tris base, 0.5% SDS, 0.5 mM EDTA) for 1.5 h at RT. The proteins were concentrated using 10 MWCO protein concentrators according to the manufacturer's protocol. Thereafter, excess unbound NEM was removed using Zeba Spin columns. Next, the samples were reduced with 20 mM DTT and incubated for 45 min at 32 °C. Afterwards, the proteins were concentrated and desalted. In the second alkylation step, the samples were incubated with d5-NEM for 1.5 h under denaturing conditions. The proteins were concentrated and washed 3× using 0.1 M Tris base, pH 7. Finally, the concentrated proteins were run on nonreducing SDS-polyacrylamide gels and stained with Coomassie Brilliant Blue.

Gel pieces were digested according to the protocol described in Shevchenko et al.[59] but without the reduction or alkylation of cysteines. The samples were subsequently analyzed by liquid chromatography–mass spectrometry (LC-MS) using an UltiMate 3000 LC (Thermo Scientific) coupled to a Q Exactive HF mass spectrometer (Thermo Scientific). Peptides analyzed on the Q Exactive HF were directly injected into an analytical column (75 μm × 300 mm), which was self-packed with 1.9 μm Reprosil Pur-AQ C18 material (Dr. Maisch, HPLC GmbH) and separated at a flow rate of 300 nl/min for 2 h with a gradient from 3% buffer A (0.1% formic acid, 1 % acetonitrile) to 40% buffer B (0.1% formic acid, 90% acetonitrile). The MS data were acquired in data-dependent acquisition mode (DDA) with an automatic switch between a full scan and up to 15 data-dependent

MS/MS scans. To obtain optimal high-quality MS/MS spectra, some samples were additionally analyzed using the same LC settings but with a scheduled parallel reaction monitoring (PRM) mode in the MS. In this mode, a target list of $m/z$ values and corresponding retention times of identified peptides from a database search of the DDA analysis was created using Skyline software. These target peptides were then selectively fragmented in the mass spectrometer. Database searches of all data were carried out with MaxQuant version 1.5.3.8[60] using the default settings; the data were searched against a *Homo sapiens*-specific database extracted from UniProt (UniProt Consortium). Cysteine plus NEM and d5-NEM, oxidation of methionine and acetylation of protein N-termini were set as the variable modifications. The results were filtered for a 1% false discovery rate (FDR) for the peptide spectrum match (PSM) and protein level[61]. For quantification, the MaxQuant results were imported into Skyline version 3.6.1[62], and extracted-ion chromatograms (EICs) were created after manually validating the automatic peak-picking.

**Differential alkylation and mmPEG$_{24}$**. Differential alkylation was performed as mentioned above except that 20 mM mmPEG$_{24}$ was used in the second alkylation step. For this procedure, cells were treated with 1 mM H$_2$O$_2$ for 30 min in RPMI medium supplemented with 0.2% FCS. The cells were then washed and lyzed, and the cytosolic fraction was collected.

For further testing of the oxidation of LPL, recombinant proteins were directly used for assays or were reduced using DTT prior to differential alkylation. Thereafter, the proteins were kept untreated or were treated with H$_2$O$_2$ for 15 min. Residual H$_2$O$_2$ was removed using 0.5 U of CAT. Samples were then subjected to differential alkylation. Following differential alkylation, the samples were run on 6% SDS-polyacrylamide gels immunoblotted for LPL (1:500).

**Western blot analysis**. Equal amounts of lysates or recombinant proteins were run on SDS-polyacrylamide gels and transferred to PVDF membranes. Thereafter, membranes were stained for indicated primary antibodies followed by staining with IRDye 680 or IRDye 800-labeled secondary antibodies (1:10,000). The membranes were scanned with LI-COR Odyssey scanner (LI-COR Biosciences)[28]. Uncropped and unprocessed scans were provided in the source data file.

**In vitro TRX1 trapping assay**. Wt and trapping mutant of TRX1-SBP 6xHis recombinant proteins were purified from M15 E. coli. Briefly, overnight-grown bacteria were IPTG-induced for 4 h. The bacteria were then pelleted by centrifugation at $5000 \times g$ for 10 min. Next, 1 L IPTG-induced bacteria were lyzed using 20 ml B-PER lysis buffer containing 10 mM imidazole for 10 min at RT. Next, the soluble fraction was collected by centrifugation at 20,000 for 30 min. Then, 1.5 ml of Ni-NTA beads was added to the soluble fraction and the samples were incubated on a rotator for 1.5 h at 4 ℃. Thereafter, samples were transferred to 5 ml polypropylene columns and washed five times using wash buffer (50 mM Na$_2$HPO$_4$, 300 mM NACI, 20 mM imidazole). Following this, the Ni-NTA beads and TRX1-SBP-6xHis complexes were incubated in 5 ml elution buffer (50 mM NaH$_2$PO$_4$, 300 mM NACI, 250 mM imidazole) for 5 min. Next, the flow-through containing purified TRX1-SBP-6xHis proteins was collected[30]. The purified recombinant proteins were then incubated with streptavidin high-sepharose beads for 1.5 h at 4 ℃. Meanwhile, $5 \times 10^6$ cells/sample were treated with the indicated concentrations of H$_2$O$_2$ or kept untreated in PBS for 5 min. Next, intracellular thiol reactions were blocked using 100 mM NEM for 5 min at RT. Unbound NEM was removed by extensive washing in PBS. The samples were then lyzed in lysis buffer (1% Triton X-100, protease inhibitor cocktail, NaVO$_4$, NAF) at a density of $1 \times 10^6$ cells/ml for 30 min on ice. The lysates were then coincubated with streptavidin-loaded recombinant-TRX1 on a rotator for 1.5 h at 4 ℃. To stop the trapping reaction, samples were incubated with 20 mM NEM for 6 min on ice. The samples were then washed using the following buffers consecutively: wash buffer 1 (1% Triton X-100, 500 mM NaCl, 1 mM NEM, 1 M Urea in 1 x TBS), wash buffer 2 (1% Triton X-100, 1 mM NEM in 1 x TBS), and wash buffer 3 (0,1% Triton X-100 in 1 x TBS). ThenTRX1-bound streptavidin beads were incubated with 5 mM biotin on a rotator for 30 min at 4 ℃. The eluates were collected by centrifugation over minispin columns and were concentrated using protein concentrators (10,000 MWCO). Then concentrated samples were divided into two fractions and mixed with reducing or nonreducing sample buffer. Finally, the samples were run on SDS-polyacrylamide gels and immunoblotted for the proteins of interest.

To test the oxidation of LPL under physiological conditions, human PBTs and PC3 cells were treated as follows: (I) human PBTs or PC3 cells were treated with 1-100 μM H$_2$O$_2$; (II) PC3 cells were treated with 5 μM and 25 μM auranofin for 10 min; and (III) PC3 cells were γ-irradiated using 30 Gy and 60 Gy prior to TRX1 trapping. γ-Irradiation was performed using a BIOBEAM GT 3000 according to the manufacturer's instructions.

**Actin bundling and sedimentation assays**. Purification of recombinant LPL protein was performed by the EMBL Protein Purification Core Facility. Recombinant LPL (50 mM Tris, 50 mM KCl, 1 mM MgCl$_2$, pH 7.2) was treated with 50 mM DTT for 45 min at RT prior to the tests. For the oxidation tests, LPL was treated with 50 μM or 200 μM H$_2$O$_2$ for 30 min at RT. Excess H$_2$O$_2$ was removed by incubating the suspensions with 0.5 U CAT. Successful excess H$_2$O$_2$ removal

was further confirmed by performing a H$_2$O$_2$ detection assay according to the manufacturer's protocol. Meanwhile, rabbit skeletal actin was dissolved in 2 mM Tris pH 7.5, 0.2 mM CaCl$_2$, and 0.2 mM ATP and incubated on ice for 1 h. The aggregates were removed by centrifugation at $14,000 \times g$ for 15 min. For the polymerization of F-actin, a 1:10 volume 10X actin polymerization buffer (500 mM KCl, 20 mM MgCl$_2$, 10 mM ATP) was added. In the final step, 5 μM actin and 5 μM LPL were mixed and incubated at RT for 4 h. The bundles were separated by centrifugation at $14,000 \times g$ for 20 min. The samples were run on 10% SDS-polyacrylamide gels and stained with Coomassie Brilliant Blue. Scanning and quantification of the signals were performed using a LI-COR Odyssey scanner and Image Studio software (LI-COR, version 3.1), respectively.

To separate G-actin from F-actin and actin bundles, polymerization of F-actin and bundle formation were performed as described above. The fractions of F-actin and F-actin bundles were then separated from G-actin by centrifugation at $150,000 \times g$ for 1 h. The pellet and supernatant fractions were collected, run on SDS-polyacrylamide gels, and processed as described above.

**Transmission electron microscopy**. To visualize actin filaments and bundles, the reaction mixtures were negatively stained. For negative staining, a glow-discharged carbon-coated formvar grid was placed on a 20 μl drop of sample and allowed to adsorb to the carbon for 10 s. The sample was then washed three times with water, stained with uranyl acetate (3% w/v), and dried. Micrographs were recorded using an electron microscope (JEM1400; Jeol Ltd) with a bottom-mounted high-sensitivity 4 K CMOS camera (TEMCAM 416; TVIPS, Gauting, Germany).

**2D migration assay and invasion assay**. To assess cell migration and invasion, 30,000 LPL eGFP- or eGFP-expressing MV3 cells in RPMI (+0.2 % FCS) were allowed to transmigrate through 8 μm pore-sized Transwell inserts into a lower compartment containing RPMI medium (+2% FCS). H$_2$O$_2$ treatments were performed in both compartments 30 min after cell seeding and cells were allowed to migrate 2.5 h in the presence or absence of H$_2$O$_2$. Then, the inserts were removed, and the cells from the upper membrane surface were wiped off with a cotton swab. The filters were then washed, fixed, and mounted on glass slides. Cells that had migrated to the collagen-coated lower side of the filter were detected by confocal microscopy. For quantification, cells in five defined optical fields were counted for each filter. The time points are indicated in the figure legends.

**Cell viability and apoptosis**. Cells were washed once with PBS and stained for Annexin-FITC and 7-AAD according to the manufacturer's instructions. The samples were directly measured by multispectral flow cytometry (LSRII, BD Biosciences).

**Immunocytochemistry**. Thirty thousand MV3 cells stably expressing eGFP, wt LPL eGFP, or LPL eGFP Cys-Ala mutants were added to either poly-D-lysine- or gelatin-coated 10 mm coverslips in 48-well plates and incubated for the indicated times. Thereafter, the cells were treated with the indicated concentrations of H$_2$O$_2$. The samples were fixed using 1.5% paraformaldehyde (PFA) for 10 min at RT. Then samples were washed and blocked in FW buffer for 30 min at RT[63]. For immunostaining, the cells were stained as indicated with the primary antibodies against LPL (1:1,000), TRX1 (1 μg ml$^{-1}$), MMP2 (1:250), and TRXR1 (1:200) in FWS buffer (PBS + 1% BSA + 0.1% saponin). Thereafter, secondary antibody staining were performed using the following antibodies and reagents for 30 min at RT: anti-rabbit Cy3 (1:800,), anti-rabbit AF405 (1:200), anti-mouse AF405 (1:200), anti-mouse Cy3 (1:800), DAPI (nuclei, 1:10,000), and SiR-actin (F-actin, 500 nM). 3D-SIM images or confocal images were obtained 1 day after sample preparation with a Nikon N-SIM microscope (×100 objective, NA 1.49).

**Simultaneous knockdown and overexpression assay**. PC3 cells (25,000) were seeded in DMEM + 1% FCS on 24-well plates and cultured overnight. The next day, the cells were transfected with one of three siRNAs targeting the 3′-UTR of LPL, one siRNA targeting the coding sequence (CDS) of LPL or a control siRNA using Lipofectamine according to the manufacturer's instructions. The next day, the medium was exchanged. Six hours later, the cells were transduced with lentiviral constructs (eGFP, wt LPL eGFP, or C101A LPL eGFP). Twenty-four hour later, medium was exchanged, and cells were cultured for up to 72 h. Western blotting and flow cytometry were used to validate knockdown of endogenous LPL and overexpression of eGFP-tagged LPL.

**Cell spreading and filopodia formation**. Twenty thousand cells were allowed to adhere to poly D-lysine-coated 10 mm coverslips in a 48-well plate. Thirty minutes after adherence, the cells were either kept untreated or were treated with 25 μM H$_2$O$_2$ and incubated for up to 3 h. Next, the samples were fixed and stained as described above. For immunostaining, the following antibodies/reagents were used: rabbit anti-MYO10 (1:250), anti-rabbit Cy3 (1:800), DAPI (1:10,000), and SiR-actin (200 nM). Images were acquired at 10 xy dimensions per sample in each experiment using confocal microscope (×100 objective, NA 1.49).

Cell spreading was analyzed by automated calculation of the cell area using the region of interest (ROI) function. For this purpose, cells were automatically

selected based on the mean pixel intensities (MPIs) of the eGFP and SiR-actin signals. All single cells with an LPL eGFP or eGFP MPI signal two-fold higher than the background signal was considered LPL eGFP-positive or eGFP-positive and processed further.

The number of filopodia in each cell was determined. Briefly, intensity profile lines were drawn at the perimeters of single cells at filopodia tips where MYO10 was enriched. Based on the intensity profiles, which specified linear sections in the images, a graph showing the MPIs was created. Next, the MPI of the background signal in each image was calculated. Then, a threshold was defined based on the background signal; i.e., the peaks on the MPI graph were counted if the MPI of the MYO10 signal at the tips of filopodia was at least 2-fold higher than that of the background. The MPI of the SiR-actin signal was also monitored in each peak for accurate quantification. Using these settings, 30 single cells were analyzed per sample.

**Gelatin matrix degradation assay.** A Cy3-labeled gelatin matrix was prepared on 10 mm coverslips according to the manufacturer's instructions. Thereafter, overnight-starved MV3 cells expressing various LPL constructs were seeded onto the matrix. Forty-five minutes after cell seeding, 25 μM $H_2O_2$ was added to the treatment groups, and the cells were cultured for 3 h in total. Next, the cells were fixed and stained for DAPI and SiR-actin as described above. Imaging was performed using a ×60 objective (NA 1.4) with a confocal microscope, and ten optical sections per sample were acquired in each experiment. Matrix degradation capacity was analyzed as the ratio between the area of faint fluorescence signal underneath a cell and the total area of the cell.

**Ratio imaging of the roGFP-Orp1 sensor.** Overnight-starved MV3 cells expressing roGFP-Orp1 or LPL-roGFP-Orp1 were measured by flow cytometry with the ratio function. Thus, eGFP and V500 spectral settings were used. In these settings, the emission wavelength is the same for both eGFP and V500, while the excitation wavelength differs. The samples were measured for 15 min. After 2 min, the indicated concentrations of $H_2O_2$ and N-acetyl cysteine (NAC) were added, and the samples were measured for up to 15 min. The time kinetics of the roGFP-Orp1 oxidation state were analyzed.

To visualize roGFP-Orp1 oxidation in the cells, a Nikon laser-scanning confocal microscope (A1) with a ratio-imaging function was used. Briefly, 30,000 cells were seeded onto 35 mm dishes overnight in RPMI medium (+0.2% FCS, without phenol red). The next day, the cells were stained with 200 nM SiR-actin for 2 h, and time-lapse imaging was performed at defined time periods. After the first time points, $H_2O_2$ or DPI were applied at the indicated concentrations. All images were acquired using a ×100 objective (NA 1.49). RoGFP was excited by the 405-nm and 488 nm laser lines and was detected by emission through a 500–554 nm bandpass filter. Simultaneously, the SiR-actin signal was measured using the confocal laser-scanning mode. Sequential scanning for roGFP (in the ratio-imaging mode) and SiR-actin (in the confocal laser-scanning mode) for each time point was performed to detect F-actin dynamics and the roGFP-Orp1 oxidation states of the cells over time. For comparison of the roGFP-Orp1 oxidation state between the cell body and filopodia, regions of interest (ROIs) were drawn in three different filopodia and three different parts of the cell body (xy positions) for each cell. The average of the ratiometric measurements of the selected ROIs was taken and used for quantification of the roGFP-Orp1 oxidation state.

**Proximity ligation assay.** MV3 cells expressing LPL were adhered to poly D-lysine-coated coverslips as described above. Thereafter, the cells were either kept untreated or were treated with 100 μM $H_2O_2$ for 5 min. Where indicated, the cells were also treated with 100 μM $H_2O_2$ or 5 μM DPI for 60 min. The cells were fixed using 1.5% PFA containing 5 mM dimedone. Next, the fixed cells were permeabilized using FWS for 15 min. Then, LPL and cysteine sulfenylation were stained using a mouse anti LPL (1:600) antibody and a rabbit anti-dimedone-specific antibody (1:1000). Next, proximity ligation reactions were initiated according to the manufacturer's instructions. Briefly, the samples were stained with anti-mouse plus and anti-rabbit minus secondary antibodies. Then, the samples were subjected to a ligation reaction for 30 min at 37 °C. The ligation was followed by PCR amplification for 90 min at 37 °C. Finally, the samples were mounted on slides using DAPI-containing mounting medium followed by 3D confocal microscopy imaging (×100, 1.49 NA). The cells were imaged with z-stacks of 0.5 μm thickness to detect all PLA puncta in the cells.

**Figures and statistics.** Statistical analyses were performed using GraphPad Prism 6.0 (GraphPad). Differences between two groups were evaluated by two-tailed $t$-tests (as indicated in the figure legends), and differences among more than two groups were evaluated by two-way ANOVA. All data are presented as the mean ± the standard error of the mean (SEM). Statistical significance was determined by the $p$-value of the statistical test, and the levels deemed as significant were as follows: $*p < 0.05$, $**p < 0.01$ $***p < 0.001$, and $****p < 0.0001$.

**Reporting summary.** Further information on research design is available in the Nature Research Reporting Summary linked to this article.

## Data availability

The authors declare that the data that support the findings of this study are available within the paper and its supplementary information files. Extra data or information are available from the corresponding authors upon request. The raw data underlying Figs. 1–8 and Supplementary Figs. 1–6 are available as Source Data.

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

## Acknowledgements

The authors thank the EMBL Protein Expression and Purification Core Facility for production of the recombinant LPL proteins. We thank PD. Dr. Tobias Dick and Dr. Benjamin Stottmeier for providing the roGFP-Orp1 and TRX1 trapping mutant constructs and for their methodological help. We thank Dr. Bernd Hessling for finalization of the mass spectrometry experiments. We thank Prof. Dr. Luise Krauth-Siegel for valuable contributions and critical discussions during the thesis advisory committee meetings. This work was supported by grants from the German Research Foundation (DFG SA393/3-4, SFB CRC156/BO4, and INST 114089/31-1 FUGG to Y.S.) and the IKTZ (S.M.).

## Author contributions

Conceptualization: E.B. and Y.S.; methodology: E.B., GH.W., B.J., C.O., R.H., and S.H.; investigation: E.B., R.H., J.L., and H.K.; resources: Y.S.; drafting of the article: E.B. and Y. S.; critical revision of the article: E.B., GH.W., K.H., and Y.S.; supervision: Y.S.; funding acquisition: Y.S., and S.M. All authors have approved the final version of the manuscript.

## Additional information

**Competing interests:** The authors declare no competing interests.

