## [Peer Review File · Nature Communications]

Reviewers' comments:

Reviewer #1 (Remarks to the Author):

This paper describes a novel observation that L-plastin is oxidized at the cell periphery, like filopodia. The oxidation occurs a specific cysteine residue, Cys101, and forms disulfide with another cysteine residue, Cys 42. The oxidized L-plastin loses its ability to induce actin bundling. In particular, it is impressive that the microscopic image analyses using various probes reveal a localized oxidation of LPL by H₂O₂ in the subcellular compartment. However, a more direct evidence for the mechanism of how the localized LPL oxidation actually correlates with actin polymerization during protrusion in the cell periphery is missing. Although the known function of LPL that stabilizes the actin polymerization was demonstrated *in vitro* using recombinant proteins, it was not verified so well in cellular system. For example, the oxidation-resistant C101A mutant partially rescues the H₂O₂-induced cell migration and invasion, but it even marginally rescues the MMP release under H₂O₂ treatment. This discrepancy might be due to the use of overexpression system for cellular study and thus weaken their conclusion. In addition, given that migration and invasion are entirely different processes, it seems awkward to link the LPL oxidation to both migration and invasion.

Major comments:

1. MV3 or PC3 cancer cells were mainly used for cellular study. MV3 melanoma cancer cells do not express LPL, while PC3 prostate cancer cells express it. It is important that cell system should be synchronized. There is a possibility that the LPL-deficient cells like MV3 are well adapted to the LPL absence. Since the localized oxidation of LPL is a key finding of this study, the LPL-dependent actin polymerization should be demonstrated by rescued experiment with the siRNA-resistant LPL expression in the cancer cell lines, i.e. PC3, depleted of endogenous LPL. It is actually important to link the LPL oxidation with actin polymerization in the peripheral protrusion.

2. Introducing MMP2 as a mechanism of LPL-dependent matrix degradation is not suitable for supporting the role of LPL oxidation. It is even hard to appreciate the significance of intracellular interaction between LPL and MMP2. Therefore, as suggested above, the cellular and *in vivo* analyses of H₂O₂-induced actin polymerization using the WT and C101A mutant-expressing cancer cells should be performed to strengthen the paper.

3. In Fig.2, data presentation is not appropriate for general audience. No detailed descriptions for ID 131, colored lines, peptide 97-123 in figure legend but 97-131 in panel a?? Therefore, the MS/MS spectra must be shown in panel a as m/z for the fragmented peptide (peptide 97 – 123) with increment of NEM or NEM-D5. Also, peptide containing Cys 42 must be shown together. What was the mean value in panel b derived from? The abundance of peptide in MS spectrum is not generally quantitative term.

In Fig 2d, what does the number of 3 (3) stand for? Frequency of what? Please modify the figure and describe the details in the figure legend.

4. In Fig. 6b, Orp1 acts as a H₂O₂ sensor and redox relay, so that it is basally in reduced state and then oxidized by H₂O₂. The oxidized Orp1 relays its oxidation state to the roGFP2 by thiol-disulfide exchange, which in turn exhibits radiometric fluorescence change. So, the drawing is wrong.

In Fig. 6d, it is interesting data showing the localized H₂O₂ at basal condition. Although the Trx1 localization was addressed, a local H₂O₂ in filopodia is known to be generated by NADPH oxidase (Ushio-Fukai, Science Signaling, 2006). Thus, this data should be supported by the knockdown of NOX isoform expressed in MV3 cells

5. Fig. 7b and c are confusing. The authors mention that expression of WT and C101A LPL increase migration and invasion in the absence of H₂O₂. If it is correct that LPL is locally oxidized as shown

in Fig 6, the C101A mutant should increase migration more strongly than WT does. Provide this evidence.

In Fig. 7d, no data for C42A LPL is provided.

6. Fig. 8 is hard to be understood. What is the difference between Fig. 1a and b? It seems that the LPL expression overall increases MMP2 activity and release. However, in Fig. 8c, the H₂O₂-induced MMP release was same between WT and C101A ($P = 0.3543$, which is different from the figure legend indicating $*=P<0.05$). Why then?

In Fig. 8d, what is the mechanism for co-localization of LPL and MMP2, latter of which is secreted and then degrades the matrix proteins? Is it just co-localization before the release? Explain.

In Fig. 8g, MMP2 level is decreased by H₂O₂. Why did not the C101A mutant rescue the MMP2 in the presence of H₂O₂? Explain.

Minor:

1. In Fig. 3c, only one blot exists. What is upper blot? May be upper band? Then, indicate it with arrow. In Fig. 3d, what is a higher MW band? Then, indicate it with arrow.

2. In Fig. 4c, Explain why two bands are detected by Prx1 antibody.

3. In Fig. 5 c and e, the black/white image is confusing with immunoblot. Just show the coomassie-stained original images.

4. In page 18, Wt or C101A LPL eGFP expressing MV3 cells were first allowed ~ ~. There is no C101A LPL data.

5. Grammatical and typing errors seen throughout the paper should be corrected. Some figure legends made hard to understand the corresponding figures.

Reviewer #2 (Remarks to the Author):

The present study investigated the regulatory mechanism of L-plastin under oxidative condition and its relevance with the dynamics of actin filament in tumor cells. The authors found that L-plastin can be oxidized at Cys42 and Cys101 residues when cells were stimulated with sub-lethal dose of H₂O₂. The oxidation level of L-plastin upon H₂O₂ treatment was enhanced in PC3 cells and MV3 cells with deficiency of Thioredoxin 1 (Trx1). In addition, L-plastin was capable to interact with recombinant Trx1, which suggests that the redox status of L-plastin was regulated by Trx1. Moreover, authors found that oxidation of L-plastin was preferentially occurred at distal protrusions of cells, where the level of Trx1 was relatively low. Functionally, oxidation of L-plastin impaired its actin bundling activity, whereas the redox-insensitive L-plastin (C101A) retained the actin bundling activity under oxidative condition. Accordingly, the mutant L-plastin partially retained migration, matrix digestion, and MMP2 release of MV3 cells upon H₂O₂ treatment. Based on these results, authors claimed that two specific Cysteine residues of L-plastin serve as redox sensors, which affect the actin-bundling activity of L-plastin under oxidative condition, leading to the altered functions of cancer cells.

Major Concerns:

1. Using differential alkylation methods and mass spectrometry, authors showed the L-plastin is oxidized at Cys42 and Cys101 (Fig1 and Fig2). Although the results were clear and novel, the oxidative regulation of a protein has been studied extensively in the field of cancer biology. For example, the oxidative regulation of Nrf2-Keap1, the master regulators of cancer metabolism, has been established for a long time (see Dinkova-Kostova et al. PNAS, 99(18):11908). Oxidation of specific Cys residues in Keap1 induces structural and functional change of Keap1, leading to the dissociation of Keap1-Nrf2 complex and subsequent Nrf2-mediated gene expression. In addition to

Keap1, there are a number of cancer-related proteins such as p53 and Ask1, which are also regulated by oxidation at Cysteine residues. Even in the field of Actin dynamics, it has been shown that F-actin assembly could be directly regulated by redox signaling via Mical (Science, 334(6063):1710). Given these previous studies, the concept that L-plastin can be regulated by oxidation is not novel, and the impact of this study to the field of cancer biology as well as redox biology is limited.

2. The role of oxidized L-plastin in the Actin dynamics and its relevance with the functions of cancer cells was unclear. For example, authors claimed that L-plastin-enhanced release of MMPs is negatively influenced by its oxidation in MV3 cells treated with H₂O₂ (Fig 8). However, there was no difference in MMPs secretion between MV3 cells expressing L-plastin and C101A L-plastin (Fig 8b-c). In Fig 8g, the difference of MMP2 secretion between WT cells and C101A L-plastin expressing cells was very small, suggesting that the biological consequence of L-plastin oxidation was either very limited or even negligible. In addition, authors used super resolution microscope to examine the colocalization of L-plastin and MMP2 (Fig 8d). However, there was no quantitative analysis, and it appears that a large population of MMP2 only localized on F-actin, but not colocalized with L-plastin. In addition to MMP release, the effect of C101A L-plastin to the migration capacity and matrix capacity in MV3 cells (Fig 7) was also very moderate. Overall, the results presented in Fig 7–8 were too preliminary to support authors' hypothesis that oxidation of L-plastin plays a role in the functions of cancer cells.

3. The current study relay solely on H₂O₂ treatment to examine the effect of oxidation on L-plastin. H₂O₂ treatment in culture cells is highly artificial, which does not necessarily represent the in vivo environment. For example, authors examined L-plastin oxidation in primary T-cells or cancer cells with 0.1 mM~1 mM H₂O₂ (Fig 1~4). It is difficult to imagine that these cells will experience such high dose of H₂O₂ in vivo. Authors should stimulate cells in other conditions and examine the oxidation level of C42 and C101 accordingly. For example, T-cells can be exposed to pathogens to stimulate NOX activity and produce endogenous H₂O₂. Cancer cells can be transplanted in mice, followed by radiation or chemotherapy. Subsequently, the oxidation level of L-plastin in these cells can be examined by differential alkylation or mass spectrometry.

4. Authors showed that oxidation of L-plastin impaired its actin-bundling activity. What is the molecular mechanism? Why the redox status of C101, but not C42, is important for the actin-bundling activity? These points are very critical and should be explored.

Minor points:

1. The title is overstated, and needs to be changed.

2. In Fig 6, authors used Ro-GFP and PLA method to examine the spatial distribution of L-plastin upon H₂O₂ treatment. Authors showed that there were a lot of F-actin-positive filopodia in cells treated with H₂O₂ (Fig 6a). In addition, L-plastin was able to localize at the distal filopodia, where the redox condition was pro-oxidative. Why F-actin structure was still observed in filopodia of H₂O₂-treated cells, if L-plastin was oxidized and lost its activity in filopodia. This discrepancy needs to be addressed.

3. In Fig 6e, only PLA puncta in cells expressing WT L-plastin was shown. It is needed to show PLA puncta in control cells, cells expressing mutant L-plastin with and without H₂O₂. In Fig 6f, the error bars of WT LPL and WT LPL+H₂O₂ were overlapped and n was 3, but there was still a significant difference between two groups. Authors need to show dot-plot to justify the analysis.

3. Statistical analysis used in Fig 6f, 6g, 7e, 8a, 8b, 8c, 8f was inappropriate. A t-test should be only used to compare the mean of two groups. ANOVA should be performed to examine the difference within >3 groups.

Reviewer #1 (Remarks to the Author):

This paper describes a novel observation that L-plastin is oxidized at the cell periphery, like filopodia. The oxidation occurs a specific cysteine residue, Cys101, and forms disulfide with another cysteine residue, Cys 42. The oxidized L-plastin losses its ability to induce actin bundling. In particular, it is impressive that the microscopic image analyses using various probes reveal a localized oxidation of LPL by H₂O₂ in the subcellular compartment. However, a more direct evidence for the mechanism of how the localized LPL oxidation actually correlates with actin polymerization during protrusion in the cell periphery is missing. Although the known function of LPL that stabilizes the actin polymerization was demonstrated in vitro using recombinant proteins, it was not verified so well in cellular system. For example, the oxidation-resistant C101A mutant partially rescues the H₂O₂-induced cell migration and invasion, but it even marginally rescues the MMP release under H₂O₂ treatment. This discrepancy might be due to the use of overexpression system for cellular study and thus weaken their conclusion. In addition, given that migration and invasion are entirely different processes, it seems awkward to link the LPL oxidation to both migration and invasion.

Author's general remarks

The authors are grateful for the constructive feedback and valuable suggestions of the reviewers. We have addressed the reviewer's concerns and believe that our manuscript has been substantially improved by the addition of new experimental data. Specifically, we further investigated the role of peripheral LPL oxidation on actin-based processes by simultaneous siRNA-mediated knock-down of endogenous LPL and lentivirus-mediated overexpression of *wt* or *C101A* LPL eGFP constructs in PC3 cells. The new data revealed that the H₂O₂-mediated decrease in cell spreading and filopodia formation can partially be rescued by the expression of redox-resistant *C101A* LPL. This underlines the importance of LPL oxidation on *Cys101* for tumor cell behavior. Moreover, additional biochemical approaches as well as the newly performed electron microscopy imaging provide further evidence that the actin-bundling capacity of LPL, but not its actin-binding capacity is influenced by its oxidation. In light of these findings, we propose a molecular model of how oxidation of LPL regulates its function on actin remodeling. In addition, we show that γ -irradiation of tumor cells induces LPL oxidation. This finding broadens the knowledge about its potential relevance to cancer therapies. Thus, our data provide important evidence for a physiological and pathophysiological significance of LPL oxidation for tumor cell behaviour.

Altogether, we now provide more insights into the molecular mechanisms of how spatial oxidation of LPL may regulate peripheral actin dynamics and tumor cell function. The specific changes in the revised manuscript are marked in red and indicated in the point-by-point response below.

Major comments:

Reviewer's comment 1

MV3 or PC3 cancer cells were mainly used for cellular study. MV3 melanoma cancer cells do not express LPL, while PC3 prostate cancer cells express it. It is important that cell system should be synchronized. There is a possibility that the LPL-deficient cells like MV3 are well adapted to the LPL absence. Since the localized oxidation of LPL is a key finding of this study, the LPL-dependent actin polymerization should be demonstrated by rescued experiment with the siRNA-resistant LPL expression in the cancer cell lines, i.e. PC3, depleted of endogenous LPL. It is actually important to link the LPL oxidation with actin polymerization in the peripheral protrusion.

Author's response 1

We thank the reviewer for addressing this point. Initially, we indeed performed the functional assays only in MV3 cells after overexpression of LPL, while PC3 cells were not used for functional tests.

As suggested by the reviewer, we have now knocked-down LPL in PC3 cells using siRNAs targeting the 3'UTR of LPL and achieved 80 % knock-down (new Fig. 9a). In order to express *wt* or mutant LPL eGFP, cells were transduced with lentiviral particles containing the respective cDNA constructs one day after siRNA transfection and LPL expression was analyzed two days post-transduction (new Fig. 9b). We then took a closer look at cell spreading and filopodia formation as measures of peripheral actin dynamics. For this purpose, unlike before, we have treated the cells with H₂O₂ for 30 min after seeding them on coverslips in order to observe the effects of LPL oxidation on the formation of adhesive structures of the cell (similar conditions as used for matrix degradation and migration assays). Measuring the cell area as a morphological parameter for cell spreading unambiguously showed that C101A LPL but not *wt* LPL expression restored cellular spreading under oxidative pressure (new Fig. 9c-e). Consistently, the number of filopodia per cell after H₂O₂ treatment was significantly diminished in *wt* LPL expressing cells while C101A LPL expressing cells were still able to form these structures (new Fig. 9f). Altogether, our data provide additional proof that spatial oxidation of LPL at Cys101 in the cell periphery is critical for actin dynamics and eventual cellular functions. These findings are also summarized in our model (new Fig. 10) and clarified in the discussion section.

Reviewer's comment 2

Introducing MMP2 as a mechanism of LPL-dependent matrix degradation is not suitable for supporting the role of LPL oxidation. It is even hard to appreciate the significance of intracellular interaction between LPL and MMP2. Therefore, as suggested above, the cellular and in vivo analyses of H₂O₂-induced actin polymerization using the WT and C101A mutant-expressing cancer cells should be performed to strengthen the paper.

Author's response 2

We agree that although expression of LPL clearly increased MMP2 release and both proteins colocalized within the cells, the influence of redox regulation of LPL on MMP2 release was marginal. We took this point of the reviewer seriously and toned down our interpretations of a link between MMP2 release and LPL oxidation. We now have placed this point to the supplement (see Fig. S6 and Author's response 6). The experiments suggested by the reviewer were performed (see Author's response 1). Thereby, we focused on investigations of peripheral LPL oxidation and actin dynamics (see Fig. 9a-f).

In addition, the newly performed biochemical assays and electron microscopy imaging provides further evidence that the actin-bundling capacity of LPL, but not its actin-binding capacity is influenced by oxidation of Cys101 (new Fig. 5f and new Fig. S3h). In light of these findings, we propose a molecular model of how oxidation of LPL regulates its function on actin remodelling (new Fig. 10).

We would like to kindly emphasize that we followed matrix degradation aspects mostly due to the observed strong rescuing effect of the C101A LPL expression on matrix degradation by tumor cells under pro-oxidative conditions (former Fig. 7e, see new Fig. 8d-e).

Reviewer's comment 3

In Fig.2, data presentation is not appropriate for general audience. No detailed descriptions for ID 131, colored lines, peptide 97-123 in figure legend but 97-131 in panel a??

Therefore, the MS/MS spectra must be shown in panel a as m/z for the fragmented peptide (peptide 97 – 123) with increment of NEM or NEM-D5. Also, peptide containing Cys 42 must be shown together.

What was the mean value in panel b derived from? The abundance of peptide in MS spectrum is not generally quantitative term.

In Fig 2d, what does the number of 3 (3) stand for? Frequency of what? Please modify the figure and describe the details in the figure legend.

Author's response 3

We apologize for the misleading explanations and have corrected the mentioned discrepancy. We had identified both peptides (aa 97-131 and aa 97-123). We now show the MS1 and MS2 chromatograms corresponding to the peptide aa 97-131. The colored lines in the former version represented naturally occurring isotopes of the same peptide. Now we present the summed intensity of all isotopes (new Fig. 2a). Moreover, *the MS/MS spectra are* now shown as m/z of the fragmented peptide and the distinctly labelled ions are indicated by different colors (new Fig. 2c). The corresponding data for Cys42 are now also included (new Fig. 2b and Fig. 2d).

The calculation of "% oxidation" (Fig. 2e) and "% sulfonylation" (Fig. 2f) is now more clearly explained in the new figure legend: Briefly, we calculate the percentage of heavy isotope-labelled peptide (d5-NEM) by dividing its intensity by that of all other forms of this peptide (light isotope labelled, unlabelled (if it exists) and hyperoxidized (SO₂H, SO₃H)). Specifically, when fragmentation of the precursor ion occurs (ms/ms

fragmentation) and thus the peptide is identified, the intensity of the precursor ion (MS1 level) is used to assess the oxidation. This relative quantification is widely used in redox biochemistry (Leichert L. et al. PNAS, 2008).

The numbers “3 (3)” indicates how often the respective peptide was identified out of the three replicate experiments (n = 3). This is now explained in the figure legend and both MS1 and MS2-level of identification is shown (see Supplementary Fig. S1c).

Reviewer’s comment 4

In Fig. 6b, Orp1 acts as a H₂O₂ sensor and redox relay, so that it is basally in reduced state and then oxidized by H₂O₂. The oxidized Orp1 relays its oxidation state to the roGFP2 by thiol-disulfide exchange, which in turn exhibits radiometric fluorescence change. So, the drawing is wrong.

In Fig. 6d, it is interesting data showing the localized H₂O₂ at basal condition. Although the Trx1 localization was addressed, a local H₂O₂ in filopodia is known to be generated by NADPH oxidase (Ushio-Fukai, Science Signaling, 2006). Thus, this data should be supported by the knockdown of NOX isoform expressed in MV3 cells

Author’s response 4

We thank the reviewer for pointing out the misleading drawing in the former Fig. 6b which we have corrected accordingly (new Fig. 6c).

As suggested by the reviewer, we analyzed expression and localization of NOX isoforms in MV3 cells which express DUOX2 and NOX4 at low levels. The latter is the predominant isoform and is considered to be the major source of endogenous ROS in these cells (Ribeiro-Pereira C. et al, Plos One, 2014). Confocal microscopy revealed that both proteins were cytoplasmic and not detectable in filopodial extensions. At basal conditions both DUOX2 and NOX4 were expressed only at low levels. Despite trying different methods, the knock-down efficiency remained marginal (data not shown). Therefore, to still address this important point, we globally inhibited the activity of NADPH oxidases with a well-established inhibitor, i.e. DPI. As shown by confocal microscopy imaging, this inhibitor slightly diminished the oxidation state of roGFP-Orp1 in the cell body of MV3 cells (Supplementary Fig. S4e), but not in filopodial extensions (Supplementary Fig. S4f) which is in line with the observed lack of NOX4 in filopodia. Apart from this, DPI treatment resulted in slightly lower levels of spatial *wt* LPL oxidation (Dimedone-PLA) (new Fig. 7d). This difference was not observed for C101A LPL, suggesting that NADPH oxidase activity in tumors might contribute to basal level of LPL oxidation at Cys101. We have extended our discussion by including these aspects and key references (Kaplan et al. *Free Radical Research*, 2011, Ushio-Fukai et al., *Science Signalling*, 2006).

Reviewer’s comment 5

Fig. 7b and c are confusing. The authors mention that expression of WT and C101A LPL increase migration and invasion in the absence of H₂O₂. If it is correct that LPL is locally oxidized as shown in Fig 6, the C101A mutant should increase migration

more strongly than WT does. Provide this evidence. In Fig. 7d, no data for C42A LPL is provided.

Author's response 5

In the absence of H₂O₂ (Supplementary Fig. S5a-b), the *wt* LPL and C101A LPL expressing cells migrate and invade at similar levels. Diminished migration by oxidation of *wt* LPL would be expected in a pro-oxidative environment as often found in a tumor microenvironment (mimicked in our experiments by the presence of H₂O₂) and as shown in our data (Fig. 8b). Under these conditions, the migration capacity was partially restored by expression of the C101A mutant. However, basal levels of oxidation were too low to make such a different behaviour apparent. In line with this, *wt* and Cys-to-Ala mutants of several other redox sensitive proteins behaved similarly in the absence of exogenous ROS (e.g. Nelson K.J. et al., Journal of Biological Chemistry, 2018 and Peralta D. et al., Nature chemical biology, 2015).

The data for C42A LPL and eGFP are now provided as new Fig. 8d.

Reviewer's comment 6

- 1. Fig. 8 is hard to be understood. What is the difference between Fig. 1a and b? It seems that the LPL expression overall increases MMP2 activity and release. However, in Fig. 8c, the H₂O₂-induced MMP release was same between WT and C101A ($P = 0.3543$, which is different from the figure legend indicating $*=P<0.05$).*
- 2. In Fig. 8d, what is the mechanism for co-localization of LPL and MMP2, latter of which is secreted and then degrades the matrix proteins? Is it just co-localization before the release? Explain.*
- 3. In Fig. 8g, MMP2 level is decreased by H₂O₂. Why did not the C101A mutant rescue the MMP2 in the presence of H₂O₂? Explain*

Author's response 6

- 1. We understand the reviewer's concerns, as the figure was not self-explanatory, and we want to apologize for that. We have detected MMP activity in the supernatant. The activity detected by means of Raw MFI was variable in different experiments. Therefore, even though we performed a pairwise comparison, the variability was too high (former Fig. 8a). Thus, we normalized the MMP activity in the supernatant of LPL expressing MV3 cells to the supernatant of control (MV3 untransduced) cells (former Fig. 8b). The former Fig. 8c showed release of MMPs in the absence or presence of H₂O₂ in each group. MMP activity in the supernatant of H₂O₂ treated group was normalized to the same group under untreated conditions. To prevent further confusion, we have removed/exchanged these figures. Now we show MMP activity in the absence or presence of H₂O₂ in relation*

to MV3 control (untransduced, untreated) in one figure (new Supplementary Fig. S6a). Data processing and statistical analysis are now better explained in the figure legends. Both former Fig. 8c and the new Supplementary Fig. S6a show that MMP activity is not correlated with LPL oxidation. The reason for this remains so far unclear. Since tissue inhibitors of metalloproteinase proteins (TIMPs), which inhibit the activity of MMPs, were also highly abundant in the supernatant, they could also influence the detected MMP activity and thereby hinder other effects.

2. The MMP2 signature on actin stress fibers (new Supplementary Fig. S6c) may be a sign for MMP2 transport on actin filaments. There is accumulating literature supporting this by showing correlations between actin filaments and microtubules and MMP-induced reorganization of the ECM (Schnaeker, E. M. et al. *Cancer Res*, 2004, Bilyug, N., *Biomol Concept*, 2016). Consistently, the MMP2 signature on fibers was lost when we treated the cells with cytochalasin D, a potent inhibitor of actin polymerization (former Supplementary Fig. S6b, new Supplementary Fig. S6d). Thus, MMP2 could be transported on actin stress fibers while LPL at the same time induces invasive structures which potentially increase the docking sites of MMPs to be released. Thereby, LPL expression may positively regulate MMP transport/release. As supporting result of a more specific interplay, the coimmunoprecipitation of MMP2 with LPL (former Fig. 8e, new Supplementary Fig. S6e) shows a direct or indirect interaction of both proteins. However, from a mechanistic perspective, our data are not yet sufficient to shed light on how LPL could contribute to the release of MMP2. We, therefore, placed this figure to the supplement (new Supplementary Fig. S6c).

Technical note:

Our SIM microscopy analysis (new Supplementary Fig. S6c; former Fig. 8d) shows stress fibers, where LPL and MMP2 come into close proximity. We also observed a strong colocalization to invasive extensions of the cells (data not shown). We chose to analyze stress fibers to be better able to visualize the potential interactions, due to the highly organized structure of the stress fibers (typically 0.4 μm thickness), and the relatively high-resolution of our SIM microscope (0.12 μm).

3. The release of MMP2 is dramatically diminished by H_2O_2 even in MV3 cells where LPL is not expressed (former Fig. 8g, new Supplementary Fig. S6f). Together with the only slight rescue of MMP2 release by expression of *C101A* LPL, this shows that H_2O_2 also affects other proteins. Particularly, transport of MMPs to the sites of release, formation of cellular extrusions through peripheral actin dynamics and MMP release are complex events where many proteins are involved and might be oxidized.

Minor concerns:

Reviewer's comment

In Fig. 3c, only one blot exists. What is upper blot? May be upper band? Then, indicate it with arrow. In Fig. 3d, what is a higher MW band? Then, indicate it with arrow.

Author's response

We are sorry for the confusion. We had two blots in the former version. However, due to redundancy with Fig. 3d, we removed one of them. We have corrected the text accordingly.

Binding of mmPEG₂₄ increases the molecular weight of LPL by 2.6 kDa and is indicative of oxidized cysteines on L-plastin. The higher molecular weight band is LPL, to which two molecules of mmPEG₂₄ are bound. We now show the upper and lower bands with tinted arrows. Please see the new Fig. 3c and Fig. 3d.

Reviewer's comment

In Fig. 4c. Explain why two bands are detected by Prx1 antibody.

Author's response

Two bands are an indication that two of the cysteines on Prx1 are bound by two different molecules of the Trx1 trapping mutant. Please see the reference below for similar example (<http://www.ub.uni-heidelberg.de/archiv/8923>).

Reviewer's comment

In Fig. 5 c and e, the black/white image is confusing with immunoblot. Just show the coomassie-stained original images.

Author's response

We have changed the color to conventional coomassie blue colors (new Fig. 5a). We have used an infrared scanner (licor odyssey, at 700 nm) to detect and quantify the actin and LPL signals on coomassie stained gels. Since the digital images can be shown in different colors, we had previously chosen black/white image for higher contrast. Please note that we now show the corresponding coomassie stained gel of wt LPL to be consistent with the revised version of the figure.

In the new Fig. 5 we now included also the respective data for C42A LPL.

Reviewer's comment

In page 18, Wt or C101A LPL eGFP expressing MV3 cells were first allowed~-. There is no C101A LPL data.

Author's response

We now provide these data. Please see Supplementary Fig. S6c, lower part.

Reviewer's comment

Grammatical and typing errors seen throughout the paper should be corrected. Some figure legends made hard to understand the corresponding figures.

Author's response

We are sorry for this and have improved the typing and grammatical errors. The figure legends were also revised.

Reviewer #2 (Remarks to the Author):

The present study investigated the regulatory mechanism of L-plastin under oxidative condition and its relevance with the dynamics of actin filament in tumor cells. The authors found that L-plastin can be oxidized at Cys42 and Cys101 residues when cells were stimulated with sub-lethal dose of H₂O₂. The oxidation level of L-plastin upon H₂O₂ treatment was enhanced in PC3 cells and MV3 cells with deficiency of Thioredoxin 1 (Trx1). In addition, L-plastin was capable to interact with recombinant Trx1, which suggests that the redox status of L-plastin was regulated by Trx1. Moreover, authors found that oxidation of L-plastin was preferentially occurred at distal protrusions of cells, where the level of Trx1 was relatively low. Functionally, oxidation of L-plastin impaired its actin bundling activity, whereas the redox-insensitive L-plastin (C101A) retained the actin bundling activity under oxidative condition. Accordingly, the mutant L-plastin partially retained migration, matrix digestion, and MMP2 release of MV3 cells upon H₂O₂ treatment. Based on these results, authors claimed that two specific Cysteine residues of L-plastin serve as redox sensors, which affect the actin-bundling activity of L-plastin under oxidative condition, leading to the altered functions of cancer cells.

Author's general remarks:

The authors are grateful for the constructive feedback and valuable suggestions of the reviewers. We have addressed the reviewer's concerns and believe that our manuscript has been substantially improved by the addition of new experimental data. Specifically, we further investigated the role of peripheral LPL oxidation on actin-based processes by simultaneous siRNA-mediated knock-down of endogenous LPL and lentivirus-mediated overexpression of *wt* or *C101A* LPL eGFP constructs in PC3 cells. The new data revealed that the H₂O₂-mediated decrease in cell spreading and filopodia formation can partially be rescued by the expression of redox-resistant *C101A* LPL. This underlines the importance of LPL oxidation on *Cys101* for tumor cell behavior. Moreover, additional biochemical approaches as well as the newly performed electron microscopy imaging provide further evidence that the actin-bundling capacity of LPL, but not its actin-binding capacity is influenced by its oxidation. In light of these findings, we propose a molecular model of how oxidation of LPL regulates its function on actin remodeling. In addition, we show that γ -irradiation of tumor cells induces LPL oxidation. This finding broadens the knowledge about its potential relevance to cancer therapies. Thus, our data provide important evidence for a physiological and pathophysiological significance of LPL oxidation for tumor cell behaviour.

Altogether, we now provide more insights into the molecular mechanisms of how spatial oxidation of LPL may regulate peripheral actin dynamics and tumor cell function. The specific changes in the revised manuscript are marked in red and indicated in the point-by-point response below.

Major Concerns:

Reviewer's comments 1

Using differential alkylation methods and mass spectrometry, authors showed the L-plastin is oxidized at Cys42 and Cys101 (Fig1 and Fig2). Although the results were clear and novel, the oxidative regulation of a protein has been studied extensively in the field of cancer biology. For example, the oxidative regulation of Nrf2-Keap1, the master regulators of cancer metabolism, has been established for a long time (see Dinkova-Kostova et al. PNAS, 99(18):11908)). Oxidation of specific Cys residues in Keap1 induces structural and functional change of Keap1, leading to the dissociation of Keap1-Nrf2 complex and subsequent Nrf2-mediated gene expression. In addition to Keap1, there are a number of cancer-related proteins such as p53 and Ask1, which are also regulated by oxidation at Cysteine residues. Even in the field of Actin dynamics, it has been shown that F-actin assembly could be directly regulated by redox signaling via Mical (Science, 334(6063):1710). Given these previous studies, the concept that L-plastin can be regulated by oxidation is not novel, and the impact of this study to the field of cancer biology as well as redox biology is limited.

Author's response 1

We are very sorry for the misunderstanding. We did not intend to say that we have found the first evidence for the importance of protein oxidation for the behavior of cancer cells. Still, the expression of LPL in cancer cells seems to be very important as it has been shown to enhance migration and metastasis of cancer cells (Riplinger, S. et al., Mol Cancer, 2014). Also, LPL is ectopically expressed in tumor cells and serves as a tumor marker which is different from all the oxidized proteins that were mentioned here. Consistently, a recent study showed that LPL is one of the top five S-thiolallylated proteins in Jurkat lymphoma cells which also suggests a high redox potential of LPL cysteines (Gruhlke, MCH. et al. Free Radic Biol Med, 2019).

We agree to the reviewer that methionine oxidation on actin by Mical is an important redox regulation that can be translated into cellular physiology. We have modified the discussion in the current version and included these aspects.

Still, we would like to kindly clarify the novelty of our work. Our results are novel not only due to oxidation of LPL *per se*, but – importantly – also due to its spatial oxidation at the periphery of the cells at basal state and in response to ROS. We have taken advantage of two novel approaches for in situ analysis of LPL oxidation, one of which (Dimedone Proximity Ligation Assay) was very recently developed (Tsutsumi, R. et al., Nature Communications, 2017). Even though oxidation in certain cellular subcompartments (ER, mitochondria), or in the vicinity of NADPH oxidases (e.g. Kaplan et al., Free Radic Res, 2011, Tsutsumi, R. et al., Nature Communications, 2017) were described, spatial oxidation at actin-based cellular protrusions was so far not shown. Spatial and organelle-independent regulation of LPL in physiological and pathophysiological settings (see also below) introduces an as yet unknown site-

specific concept of regulation. To experimentally substantiate this point, we now provide new evidences that LPL regulation is critical for actin dynamics at the cell periphery (see new Fig. 9a-f).

Reviewer's comments 2

The role of oxidized L-plastin in the Actin dynamics and its relevance with the functions of cancer cells was unclear. For example, authors claimed that L-plastin-enhanced release of MMPs is negatively influenced by its oxidation in MV3 cells treated with H₂O₂ (Fig 8). However, there was no difference in MMPs secretion between MV3 cells expressing L-plastin and C101A L-plastin (Fig 8b-c). In Fig 8g, the difference of MMP2 secretion between WT cells and C101A L-plastin expressing cells was very small, suggesting that the biological consequence of L-plastin oxidation was either very limited or even negligible.

In addition, authors used super resolution microscope to examine the colocalization of L-plastin and MMP2 (Fig 8d). However, there was no quantitative analysis, and it appears that a large population of MMP2 only localized on F-actin, but not colocalized with L-plastin. In addition to MMP release, the effect of C101A L-plastin to the migration capacity and matrix capacity in MV3 cells (Fig 7) was also very moderate. Overall, the results presented in Fig 7~8 were too preliminary to support authors' hypothesis that oxidation of L-plastin plays a role in the functions of cancer cells.

Author's response 2

We understand the reviewer's concerns, as Fig. 7 and Fig. 8 were not self-explanatory, and we want to apologize for that. We have performed additional experiments and exchanged some figures for clarification.

Specifically, we provide further evidence to the impact of spatial LPL oxidation at the periphery for cell spreading and filopodia formation. Initially, we performed the functional assays only in MV3 cells after overexpression of LPL, while PC3 cells were not used for functional tests. Now we have now knocked-down LPL in PC3 cells using siRNAs targeting the 3'UTR of LPL and achieved 80 % knock-down (new Fig. 9a). In order to express *wt* or mutant LPL eGFP, lentiviral particles containing the respective cDNA constructs were transduced in the cells one day after siRNA transfection and LPL expression was analyzed two days post-transduction (new Fig. 9b). We then took a closer look at cell spreading and filopodia formation as measures of peripheral actin dynamics. For this purpose, unlike before, we have treated the cells with H₂O₂ for 30 min after seeding them on coverslips in order to observe the effects of LPL oxidation on the formation of adhesive structures of the cell (similar conditions as used for matrix degradation and migration assays). Measuring the cell area as a morphological parameter for cell spreading unambiguously showed that C101A LPL but not *wt* LPL expression restored cellular spreading under oxidative pressure (new Fig. 9c-e). Consistently, the number of filopodia per cell after H₂O₂ treatment was significantly diminished in *wt* LPL expressing cells while C101A LPL expressing cells were still able to form these structures (new Fig. 9f). Altogether, our data provide additional proof that spatial oxidation of LPL at *Cys101* in the cell periphery is critical for actin dynamics

and eventual cellular functions. These findings are also summarized in our model (new Fig. 10) and clarified in the discussion section.

We actually do not understand the following statement of the reviewer:

“In addition to MMP release, the effect of C101A L-plastin to the migration capacity and matrix capacity in MV3 cells (Fig 7) was also very moderate. Overall, the results presented in Fig 7~8 were too preliminary to support authors’ hypothesis that oxidation of L-plastin plays a role in the functions of cancer cells.”

Since:

- 1) We observed a strong rescuing effect of the C101A LPL expression on matrix degradation by tumor cells under pro-oxidative conditions (former Fig. 7e, see new Fig. 8d-e, $p < 0.0001$).
- 2) There is a significant recovery of migration and invasion by expression of the C101A mutant of LPL in the presence of H_2O_2 (former Fig. 7b-c, new Fig. 8b-c). We would not expect a complete recovery of migration or invasion which are complex events in which many proteins are involved, some of which may also be oxidized.
- 3) Together with the new findings described above (new Fig. 9a-f), we see a clear evidence that oxidation of L-plastin plays a role in the functions of cancer cells.

However, we agree with the reviewer that:

“the difference of MMP2 secretion between WT cells and C101A L-plastin expressing cells was very small”.

The influence of oxidation of LPL on MMP2 release was indeed marginal (former Fig. 8g and the new Supplementary Fig. S6b). In addition, the former Fig. 8c and the new Supplementary Fig. S6a show that MMP activity is not correlated with LPL oxidation. The reason for this remains so far unclear. Since tissue inhibitors of metalloproteinase proteins (TIMPs), which inhibit the activity of MMPs, were also highly abundant in the supernatant, they could also influence the detected MMP activity and thereby hinder other effects.

SIM microscopy suggested a colocalization of MMP2 and LPL on actin stress fibers (former Fig. 8d, new Supplementary Fig. S6c). The potential interaction between MMP2 and LPL could be independently underlined by coimmunoprecipitation of both proteins (former Fig. 8e, new Supplementary Fig. S6e). MMP2 signature on actin stress fibers may be a sign for MMP2 transport on actin filaments. There is accumulating literature supporting this by showing correlations between actin filaments and microtubules and MMP-induced reorganization of the ECM (Schnaeker, E. M. et al. Cancer Res, 2004, Bilyug, N., Biomol Concept, 2016). Consistently, the MMP2 signature on fibers was lost when we treated the cells with cytochalasin D, a potent inhibitor of actin polymerization (former Supplementary Fig. S6b, new Supplementary Fig. S6d). Thus, MMP2 could be transported on actin stress fibers while LPL at the same time induces invasive structures which potentially increase the docking sites of

MMPs to be released. Thereby, LPL expression may positively regulate MMP transport/release. However, from a mechanistic perspective, our data are not yet sufficient to shed light on how LPL could contribute to the release of MMP2. We, therefore, placed this figure to the supplement.

Technical note:

Our SIM microscopy analysis (new Supplementary Fig. S6c; former Fig. 8d) shows stress fibers, where LPL and MMP2 come into close proximity. We also observed a strong colocalization to invasive extensions of the cells (data not shown). We chose to analyze stress fibers to be better able to visualize the potential interactions, due to the highly organized structure of the stress fibers (typically 0.4 μm thickness), and the relatively high-resolution of our SIM microscope (0.12 μm).

Reviewer's comments 3

The current study relay solely on H₂O₂ treatment to examine the effect of oxidation on L-plastin. H₂O₂ treatment in culture cells is highly artificial, which does not necessarily represent the in vivo environment. For example, authors examined L-plastin oxidation in primary T-cells or cancer cells with 0.1 mM~1 mM H₂O₂ (Fig 1~4). It is difficult to imagine that these cells will experience such high dose of H₂O₂ in vivo. Authors should stimulate cells in other conditions and examine the oxidation level of C42 and C101 accordingly. For example, T-cells can be exposed to pathogens to stimulate NOX activity and produce endogenous H₂O₂. Cancer cells can be transplanted in mice, followed by radiation or chemotherapy. Subsequently, the oxidation level of L-plastin in these cells can be examined by differential alkylation or mass spectrometry.

Author's response 3

We thank the reviewer for the suggestions. We now present new data that provide additional evidence for the physiological and pathophysiological relevance of LPL oxidation. Particularly, γ -irradiation of PC3 cells and MV3 cells led to oxidation of LPL (new Fig. 4i-j). This finding broadens the knowledge about its potential relevance to cancer therapies.

The treatments used for the experiments shown in Figs. 1-4 were short-term (5 min for trapping experiments and 30 min for mass spectrometry) with high concentrations of H₂O₂. The concentrations applied were in the range that is widely used in similar studies, especially for biochemical characterization of protein oxidation (Held J.M., *et al.* Molecular and cellular proteomics, 2010 and Peralta D. *et al.*, Nature chemical biology, 2015) in order to override the rapid response of the cellular antioxidant systems (Sobotto M.C. *et al.*, Free radical biology and medicine, 2013). To show that LPL can be oxidized even with very low concentrations of H₂O₂, we took advantage of Trx1 kinetic trapping. These new experiments revealed that LPL oxidation can be observed in tumor cells in response to low concentrations of H₂O₂, i.e. 5 μM (Fig. 3e). In untransformed T-cells, which have lower antioxidant capacity compared to tumor

cells, LPL is strongly oxidized with H₂O₂ concentrations as low as 1 μM (Fig. 3f). Independent from these findings, we show that spatial oxidation of LPL is diminished when NADPH oxidase activity was blocked by DPI treatment. This also suggests that oxidation of LPL can take place physiologically in the vicinity of NADPH oxidases.

Reviewer's comments 4

Authors showed that oxidation of L-plastin impaired its actin-bundling activity. What is the molecular mechanism? Why the redox status of C101, but not C42, is important for the actin-bundling activity? These points are very critical and should be explored.

Author's response 4

We now provide data on the actin-bundling capacity of C42A LPL which were missing in the former version. They clearly show a moderate rescuing effect with C42A LPL in comparison to C101A LPL (new Fig. 5e-f). However, the redox status on Cys101 is central to diminished actin-bundling capacity of LPL in a pro-oxidative milieu.

To provide mechanistic insights into these findings, in addition to actin-bundling, we now tested the actin-binding capacity of *wt* LPL, C42A or C101A LPL by use of an actin sedimentation assay (new Supplementary Fig. S3f-g). The results indicate that the amount of LPL in the actin filament fraction (pellet) is comparable for all LPL constructs. This shows that, in contrast to actin bundling, the actin-binding capacity of LPL is not influenced upon its oxidation. Consistently, new experiments using transmission electron microscopy (TEM) imaging of actin filaments/bundles showed that in a pro-oxidative environment only C101A LPL, not *wt* LPL, could form actin bundles (see Fig. 5f and Supplementary Fig. S3h).

Together, the additional biochemical approaches as well as the newly performed electron microscopy imaging provided further evidence that the actin-bundling capacity of LPL, but not its actin-binding capacity is influenced by its oxidation. In light of these findings and previous models regarding actin binding (Galkin *et al.*, PNAS, 2008) and bundling (Ishida, H. *et al.*, Scientific reports, 2017) by LPL, we propose a molecular model of how oxidation of LPL regulates its function on actin remodeling (please see in the new Fig.10a).

Minor concerns

Reviewer's comments

The title is overstated, and needs to be changed.

Author's response

We have changed the title.

Reviewer's comments

In Fig 6, authors used Ro-GFP and PLA method to examine the spatial distribution of L-platin upon H₂O₂ treatment. Authors showed that there were a lot of F-actin-positive filopodia in cells treated with H₂O₂ (Fig 6a). In addition, L-plastin was able to localize at the distal filopodia, where the redox condition was pro-oxidative. Why F-actin structure was still observed in filopodia of H₂O₂-treated cells, if L-plastin was oxidized and lost its activity in filopodia. This discrepancy needs to be addressed.

Author's response

In the former Fig. 6a, new Fig. 7a, we have kept the cells untreated or treated with H₂O₂ for 30 min. Prior to treatment, the cells were allowed to adhere on poly-D-lysine coated coverslips overnight to allow firm adherence. Thus, the majority of the filopodia has been formed already before the treatment. Additionally, our findings show that oxidation of LPL prevents formation of new actin bundles (see Author's response 4). Thus, to properly define actively formed filopodia (conventional filopodia) in response to a stimulus, and to find out the role of LPL oxidation in these processes, we performed new experiments that started H₂O₂ treatments shortly after cell adherence (See new Fig. 9d.)

Reviewer's comments

In Fig 6e, only PLA puncta in cells expressing WT L-plastin was shown. It is needed to show PLA puncta in control cells, cells expressing mutant L-plastin with and without H₂O₂.

Author's response

We now show representative images for control cells and LPL expressing cells in the presence and absence of H₂O₂ treatment (new Fig. 7a).

Reviewer's comments

In Fig 6f, the error bars of WT LPL and WT LPL+H₂O₂ were overlapped and n was 3, but there was still a significant difference between two groups. Authors need to show dot-plot to justify the analysis.

Author's response

We now show dot plots (new Fig. 7a). The reason for high error bars is the efficiency of the PLA reaction. There is variability in each experiment, however, the increase in the number of PLA puncta upon H₂O₂ treatment was still consistent.

Reviewer's comments

Statistical analysis used in Fig 6f, 6g, 7e, 8a, 8b, 8c, 8f was inappropriate. A t-test should be only used to compare the mean of two groups. ANOVA should be performed to examine the difference within >3 groups.

Author's response

We are sorry for our misleading information. We have performed ANOVA for multiple comparisons and paired/unpaired t-tests for comparing two groups. We have modified

the description of the statistical analyses in the Materials & Methods section accordingly.

Reviewers' comments:

Reviewer #1 (Remarks to the Author):

The revised manuscript contains a set of new data, which addresses most of the previously-raised queries. Through new experiments, the spatial oxidation of LPL is well characterized and now convinced at biochemical and cellular levels. However, the functional significance of LPL oxidation obtained from the two types of cancer cells still weakens the discovery of the study. It is probably because H₂O₂ can target various redox-sensitive proteins involved in migration and invasion, which in turn compromise the LPL oxidation effect. To strengthen the study, a relevant in vivo experiment, i.e. tumor metastasis assay using the LPL C101A mutant-expressing cancer cells in the mouse model, would be effective, if possible.

The following minor issues need to be fixed.

1. It is unclear whether PC3 cells naturally express LPL or ectopically established the LPL expression. Please check the dictionary meaning of ectopic (page 7, bottom).
2. Figs 1 and 2 can be combined for better understanding of the readers
3. Please clarify the cell types used in Fig 5. Supposed to be MV3 cells.
4. The sentences in the front of page 9 is confusing. For example, the sentence " Note that the actin-bundling function ~ ~ ~ MV3 cells" does not match with data shown in Fig S3f and g, wherein graph shows no difference between control and H₂O₂ treatment. I guess the sentence actually mentions the Fig 5d and e. Rather, the following sentences seem like the right description for Fig S3f and g.
5. The figure legend in the Fig S3 should be checked. The panel a seems the coomassie-stained, not immunoblot, gel image, like that in Fig 5a. The panels f and g must contain more detailed description.
6. Discussion part is too long by repeatedly describing the results (pages 15 – 17). Thus, the manuscript needs to be read by native English editors

Reviewer #2 (Remarks to the Author):

The authors have addressed most of my concerns. I have no further comment.

Point by point response revision manuscript Balta et al.

Reviewer's remarks to the author:

*The revised manuscript contains a set of new data, which addresses most of the previously-raised queries. Through new experiments, the spatial oxidation of LPL is well characterized and now convinced at biochemical and cellular levels. However, the functional significance of LPL oxidation obtained from the two types of cancer cells still weakens the discovery of the study. It is probably because H₂O₂ can target various redox-sensitive proteins involved in migration and invasion, which in turn compromise the LPL oxidation effect. To strengthen the study, a relevant *in vivo* experiment, i.e. tumor metastasis assay using the LPL C101A mutant-expressing cancer cells in the mouse model, would be effective, if possible.*

Author's remarks to the reviewer:

We thank the reviewer for pointing out the aspect of an adequate *in vivo* experiment to further consolidate our results. However, we are convinced that we cannot address this issue within a reasonable period of time. Establishing and conducting of adequate *in vivo* experiments would take an unpredictably long time. Therefore, this approach would be a risk of losing the novelty of our findings regarding spatial oxidation of LPL at Cys101 and Cys42 and its implications for actin bundling, for the formation of cell protrusions, and for cancer cell behavior. Thus, we believe that *in vivo* experiments are beyond the scope of this manuscript and are only considered for our long-term planning.

Below, we would like to briefly explain our reasons in more detail and outline our current plans to make this possible in the future.

1. Lack of an adequate prooxidative tumor micromilieu in xenograft tumor models:

The complexity of a tumor intrinsic or therapeutically induced prooxidative tumor micromilieu (for review see Hedegüs et al., Redox biology 16, pp 59-74, 2018) would necessitate establishment of novel mouse models.

a) Lack of natural immune cells

One important component contributing to a prooxidative tumor micromilieu is the immune system. However, in a mouse xenograft model, human MV3 or PC3 cells that express *wt* or C101A LPL would need to be injected in nude mice lacking an adequate immune system. Therefore, such experimental conditions would not reflect the tumor microenvironment present in tumor patients.

b) Lack of experimental methods to investigate the effects of redox regulation on the metastatic processes in mouse xenograft models:

Redox regulation is unpredictable in current studies performed with metastatic colony formation assays. In this regard, the choice of injection route and the time, dosage and means of *in vivo* ROS induction for influencing metastatic

processes where LPL oxidation is relevant requires several lines of establishments.

Therefore, we are currently concentrating on the establishment of suitable xenograft models with additional engraftment of immune cells and/or redox-modified tumor cells to investigate the importance of redox regulation.

2. Lack of knowledge about LPL oxidation in mouse cells:

Mouse models in which *C101A* and *wt* LPL are expressed in mouse tumor cells would also be only a long-term option. In this context, even though *Cys101* is conserved in mouse and human LPL, redox regulation and functional consequences of LPL oxidation in mouse cells still need to be characterized. This requires a series of biochemical and *in vitro* experiments with mouse cells before the *in vivo* experiments can be started.

3. Legal issues for animal experiments in Germany:

To be allowed to investigate the effects of LPL oxidation on tumor cell behavior *in vivo*, we would need to apply for a new license for animal experimentation. The approval of such licenses in Germany takes at least 6 months. Therefore, we are planning to perform the *in vivo* experiments as a follow-up study.

Reviewers minor comments:

Reviewer's comment 1

It is unclear whether PC3 cells naturally express LPL or ectopically established the LPL expression. Please check the dictionary meaning of ectopic (page 7, bottom).

Author's response 1

Physiologically, LPL is only expressed in cells of hematopoietic origin. Non-hematopoietic cells do not express LPL, unless they are malignantly transformed. LPL expressing PC3 prostate cancer cells originated from a cancer patient and have an epithelial origin. Therefore, we have used the term "ectopic" which is defined as: "The occurrence of gene expression in a tissue in which it is normally not expressed." This is now explained in the introduction part.

Reviewer's comment 2

Figs 1 and 2 can be combined for better understanding of the readers

Author's response 2

We have combined Fig.1 and Fig.2. To make this possible, we have moved the mass spectrometry data for *C42A* LPL to the supplementary figures (new Supplementary Fig. S1b-c). Please note that the order of the figures has changed accordingly.

Reviewer's comment 3

Please clarify the cell types used in Fig 5. Supposed to be MV3 cells.

Author's response 3

We are sorry for the confusion. Former Fig. 5 (new Fig.4) shows the biochemical characterization of the actin-bundling capacity of LPL on F-actin. This assay was performed using recombinant proteins. We have now made this point clearer in the results and figure legends.

Reviewer's comment 4

The sentences in the front of page 9 is confusing. For example, the sentence "Note that the actin-bundling function ~~~ MV3 cells" does not match with data shown in Fig S3f and g, wherein graph shows no difference between control and H2O2 treatment. I guess the sentence actually mentions the Fig 5d and e. Rather, the following sentences seem like the right description for Fig S3f and g.

Author's response 4

Again, we are very sorry for the confusion. We have meant that actin **bundling** - but not actin **binding** - is influenced by ROS. We have referred to the corresponding figures correctly.

Reviewer's comment 5

The figure legend in the Fig S3 should be checked. The panel a seems the coomassie-stained, not immunoblot, gel image, like that in Fig 5a. The panels f and g must contain more detailed description.

Author's response 5

We thank to the reviewer for this remark. We have changed the text accordingly. Supplementary Fig. S3f-g are explained in more details.

Reviewer's comment 6

Discussion part is too long by repeatedly describing the results (pages 15 – 17). Thus, the manuscript needs to be read by native English editors

Author's response 6

We thank the reviewer for the constructive suggestions. We have shortened our discussion part and removed repetitions.

Our manuscript has been edited for English language usage, grammar, spelling and punctuation by native speakers at the "Nature research editing service". The editing service is acknowledged in the acknowledgement. The respective changes are highlighted throughout the text.